# Enhanced multi-carbon alcohol electroproduction from CO via modulated hydrogen adsorption

Jun Li [1,2,7], Aoni Xu[2,3,7], Fengwang Li [2], Ziyun Wang [2], Chengqin Zou[2,4], Christine M. Gabardo [1], Yuhang Wang [2], Adnan Ozden[1], Yi Xu [1], Dae-Hyun Nam [2], Yanwei Lum [2], Joshua Wicks [2], Bin Chen[2], Zhiqiang Wang [5], Jiatang Chen [5], Yunzhou Wen [6], Taotao Zhuang[2], Mingchuan Luo[2], Xiwen Du [4], Tsun-Kong Sham[5], Bo Zhang [6], Edward H. Sargent [2✉] & David Sinton [1✉]

Multi-carbon alcohols such as ethanol are valued as fuels in view of their high energy density and ready transport. Unfortunately, the selectivity toward alcohols in $CO_2$/CO electro-reduction is diminished by ethylene production, especially when operating at high current densities (>100 mA $cm^{-2}$). Here we report a metal doping approach to tune the adsorption of hydrogen at the copper surface and thereby promote alcohol production. Using density functional theory calculations, we screen a suite of transition metal dopants and find that incorporating Pd in Cu moderates hydrogen adsorption and assists the hydrogenation of $C_2$ intermediates, providing a means to favour alcohol production and suppress ethylene. We synthesize a Pd-doped Cu catalyst that achieves a Faradaic efficiency of 40% toward alcohols and a partial current density of 277 mA $cm^{-2}$ from CO electroreduction. The activity exceeds that of prior reports by a factor of 2.

[1] Department of Mechanical and Industrial Engineering, University of Toronto, 5 King's College Road, Toronto, ON M5S 3G8, Canada. [2] Department of Electrical and Computer Engineering, University of Toronto, 10 King's College Road, Toronto, ON M5S 3G4, Canada. [3] Institute for Advanced Materials and Technology, University of Science and Technology Beijing, Beijing 100083, China. [4] Institute of New-Energy Materials, School of Materials Science and Engineering, Tianjin University, Tianjin 300072, China. [5] Department of Chemistry, University of Western Ontario, 1151 Richmond Street, London, ON N6A 5B7, Canada. [6] State Key Laboratory of Molecular Engineering of Polymers, Department of Macromolecular Science and Laboratory of Advanced Materials, Fudan University, Shanghai 200438, China. [7] These authors contributed equally: Jun Li, Aoni Xu. ✉email: ted.sargent@utoronto.ca; sinton@mie.utoronto.ca

The need to utilize carbon and to store intermittent renewable electricity has motivated electrochemical $CO_2$ reduction ($CO_2$R) to carbon-based fuels and chemicals[1–3]. Of particular interest is the electroproduction of multi-carbon products ($C_{2+}$), including gaseous ethylene[4–7] and liquid oxygenates[3,8–10] such as acetate, ethanol and propanol.

Multi-carbon alcohol (i.e. ethanol and propanol) are high in energy density and thus readily integrated into existing fuel distribution and utilization infrastructure[3,11–14]. Copper-based materials have been advanced recently to promote carbon–carbon (C–C) coupling reactions towards $C_{2+}$ products, yet the selectivity towards alcohol from $CO_2$R is poor—instead, ethylene production predominates[4,6,15].

Beginning from CO avoids $CO_2$ loss to carbonate and CO is increasingly available from industrial streams and from renewable $CO_2$R[11,16,17]. Use of CO feedstock boosts CO coverage on the Cu surface, favouring alcohols[18], leading to an impressive 60% selectivity for alcohol production at low current densities (≤1 mA $cm^{-2}$)[12]. At high current densities (~300 mA $cm^{-2}$), alcohol selectivity on Cu drops below 40%[16,17]. Given the importance of hydrogenation in steering $C_2$ formation pathways[19,20], we posited herein that a loss of adsorbed hydrogen might account for the loss of alcohol selectivity at high productivity.

Here we report a strategy in which a transition metal dopant is introduced at the Cu catalyst surface to produce local active H species and steer COR (electrochemical CO reduction) to alcohols. Computationally, we screen a library of transition metal dopants with distinct H-binding abilities, and we calculate the corresponding thermodynamic reaction energies along the alcohol pathway. We find that Pd dopants provide optimal H-binding for alcohol production at neighbouring Cu sites, hydrogenating the post-C–C coupling reaction intermediates along the alcohol pathway. We synthesize Pd-modified Cu catalysts and verify the atomistic electronic structure of Pd in Cu using X-ray absorption spectroscopy (XAS). When these catalysts are experimentally tested via COR, we achieve an alcohol selectivity of 40% and an alcohol partial current density of 277 mA $cm^{-2}$ at −0.62 V versus reversible hydrogen electrode (vs. RHE). We also prepare a Pt-modified Cu catalyst to confirm the enhancement of alcohol selectivity via the tuning of H-binding. The secondary metal dopant in bimetallic Cu catalysts modulates the adsorbed H species and promotes alcohol production over ethylene, exhibiting a twofold increase in the alcohol-to-ethylene ratio compared to bare-Cu catalysts.

## Results and discussion

**Density functional theory calculations**. CO dimerization has been suggested as the rate-determining step for CO-to-$C_{2+}$ conversion, generally[2,21,22]. Recent works by Goddard and co-workers[19,20] have shown that the reaction of the intermediate HOCCH* through either a hydrogenation pathway (Eq. 1) or a dehydroxylation pathway (Eq. 2) determines the selectivity towards alcohol vs. ethylene (Fig. 1a and Supplementary Fig. 1):

$$HOCCH^* + H^* \rightarrow CHCHOH^*, \qquad (1)$$

$$HOCCH^* + e^- \rightarrow CCH^* + OH^-. \qquad (2)$$

A similar mechanism has also been proposed by Koper and co-workers[2,22], whereby hydrogenation of $C_2$ intermediates leads to acetaldehyde and subsequently ethanol generation. We reasoned that controlling the catalytic hydrogenation of HOCCH* intermediates could steer high-rate COR selectivity from ethylene to alcohols.

The introduction of H-binding elements into host structures has been reported to provide active hydrogen adsorption, enabling selective hydrogenation of hydrocarbon species[23–25]. We postulated that integrating H-absorption active elements into the Cu matrix would provide a means to boost the hydrogenation of HOCCH* intermediates.

To validate our hypothesis, we assessed the DFT-calculated reaction free energies ($\Delta G$, Supplementary Note 1) for the hydrogenation of HOCCH* intermediates on a clean Cu surface using two distinct proton sources: adsorbed H* and H taken from water (Supplementary Fig. 2 and Supplementary Table 1). We determined that the hydrogenation of HOCCH* via a Cu-adsorbed H* ($\Delta G = 0.04$ eV) is energetically more favourable than hydrogenation via a water molecule ($\Delta G = 0.13$ eV).

We introduced a suite of transitional metal dopants with varying H-binding abilities[26], and calculated the hydrogenation reaction free energies of HOCCH* at the resulting bimetallic Cu surfaces. These calculations revealed a volcano-like relationship between the hydrogenation reaction free energies of HOCCH* and H-absorption abilities of dopants (Fig. 1b and Supplementary Tables 2–3): Strong H-binding on Cu–M (e.g. M = W (tungsten) with a $\Delta G = 0.30$ eV) fixed adsorbed H* and decelerated the hydrogenation of HOCCH*, whereas weak H-binding (e.g. M = Ir with a $\Delta G = 0.15$ eV) failed to sufficiently stabilize H. We also calculated the reaction free energies of the dehydroxylation process of HOCCH* (Supplementary Fig. 3 and Supplementary Table 4)—an intermediate key to ethylene formation[19]—and observed no effect of metal doping on the ethylene formation pathway.

The DFT results indicated that using Pd as a dopant results in the lowest reaction free energy of hydrogenation of HOCCH* ($\Delta G = -0.15$ eV) of all the screened dopants by activating absorbed H*. This finding suggests that a Pd dopant could achieve optimal H-binding at a Cu–Pd surface and readily hydrogenate HOCCH* formed on neighbouring Cu atoms and thereby increase alcohol production.

We further investigated the influence of the Pd coordination environment on alcohol selectivity by varying the doping configurations (Supplementary Fig. 4), with atomically dispersed Pd dopants and Pd aggregates. The atomic-level Pd-doped Cu configurations exhibit lower hydrogenation free energies than Cu surfaces with Pd aggregates, as well as better stabilities from surface energy calculations (Fig. 1c and Supplementary Note 2), suggesting an enhancement in alcohol selectivity at Cu catalyst surfaces with atomic Pd doping. We also assessed the effect of dopant distributions on competing reactions, the dehydroxylation of HOCCH* and the hydrogen evolution reaction (Supplementary Figs. 5–7), and found that these competing reactions become more favourable with aggregated Pd, further motivating an atomic Pd doping approach for alcohol production.

**Materials characterization**. In light of the DFT results, we synthesized Pd-doped Cu catalysts using a selective chemical etching method via galvanic replacement in the presence of Pd ions (See "Methods" for details)[27]. We characterized Cu catalysts after COR using transmission electron microscopy (TEM) and scanning electron microscopy (SEM). The morphology of the Cu nanoparticles with a mean size of ~100 nm did not change before and after Pd loading (Fig. 2a and Supplementary Fig. 8). Similar polycrystalline Cu structures for Pd-doped Cu and bare-Cu catalysts were revealed via powder X-ray diffraction (P-XRD, Supplementary Fig. 9) analysis. The Pd dopants were determined to be evenly distributed in the Cu structure using the aberration-correction high-angle annular dark-field scanning transmission electron microscopy (HAADF-STEM), coupled with electron

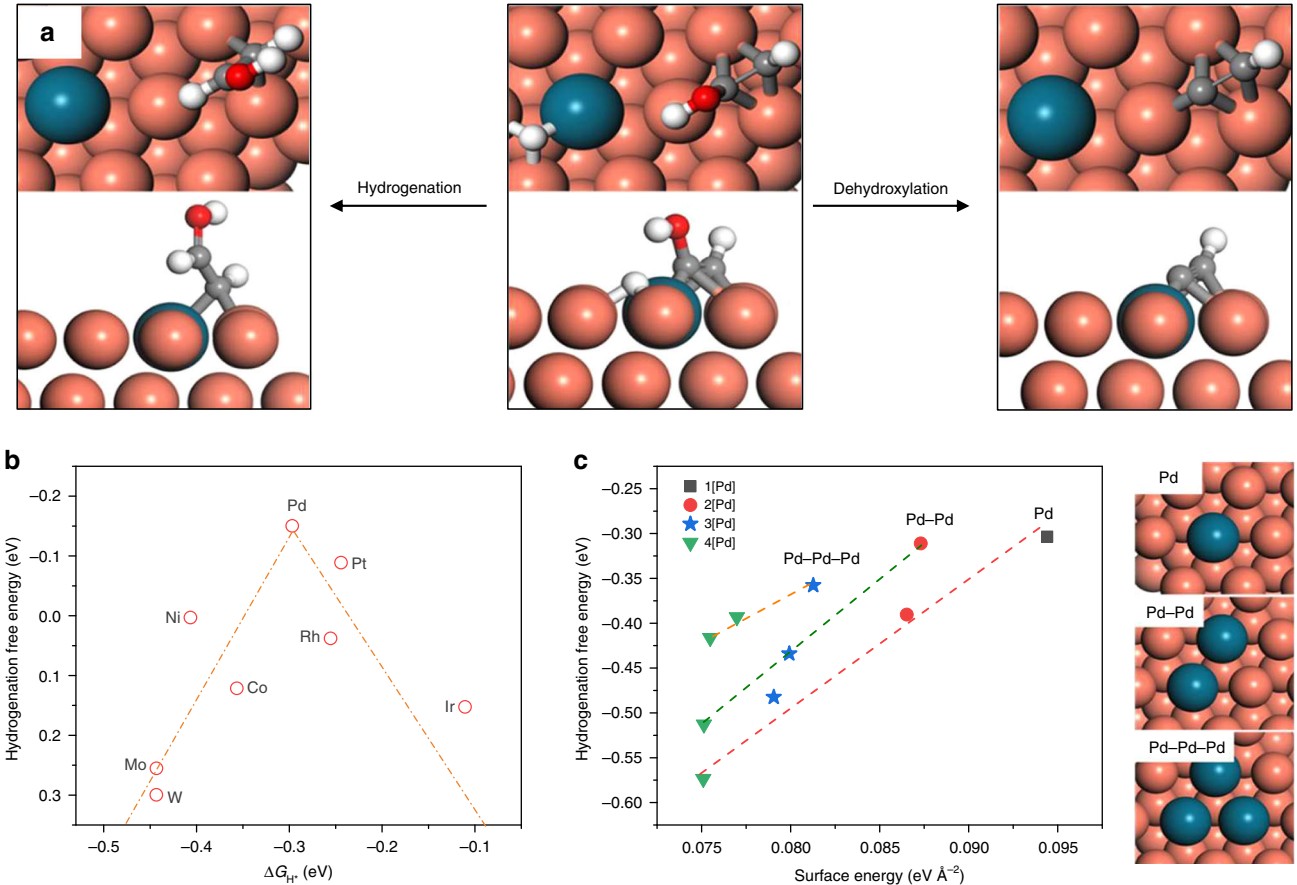

**Fig. 1 DFT reaction free energy calculations of HOCCH\* intermediates. a** A schematic view of the hydrogenation with adsorbed H\* and dehydroxylation of HOCCH\*. Red, white, grey, orange and dark blue balls represent oxygen, hydrogen, carbon, copper and dopant, respectively. **b** Plot of the calculated hydrogenation reaction free energies of HOCCH\* against the H adsorption energies of different dopants. **c** The hydrogenation reaction free energies of HOCCH\* using different Pd doping configurations. 1[Pd], 2[Pd], 3[Pd], 4[Pd] refer to Pd-modified Cu models with surface Pd concentration of 1/16, 1/8, 3/16, 1/4, respectively.

energy loss spectroscopy (EELS) mapping (Fig. 2b, c), as well as energy-dispersive X-ray (EDX) mapping (Supplementary Fig. 10). Individual Pd atoms were discerned on the crystal surface of Cu in the Pd-doped Cu catalysts based on the difference in Z-contrast. The presence of Pd at the surface of Cu nanoparticles was further verified via X-ray photoelectron spectroscopy (XPS) analysis (Fig. 2d, e). Using a Pd:Cu ratio of 0.01 in the precursor solution, a similar Pd:Cu ratio of 0.007 was achieved, as measured by inductively coupled plasma optical emission spectroscopy (ICP-OES, Supplementary Table 5).

We conducted XAS measurements to determine the electronic nature of the CuPd catalysts, including X-ray absorption near-edge structure (XANES) and extended X-ray absorption fine structure (EXAFS), at the Cu K-edge and Pd K-edge, respectively. To assess the effect of Pd loading on the catalyst structures, we varied the Pd loading in Cu with nominal Pd:Cu ratios of 0.01 and 0.015 by tuning the concentration of the Pd precursor solution during synthesis, resulting in Pd:Cu ratios of 0.007 and 0.011 determined by ICP-OES, respectively, termed $CuPd_{0.007}$ and $CuPd_{0.011}$. We performed operando XAS analysis and found that Cu species remained in the metallic state during COR (Fig. 3a and Supplementary Fig. 11). The Cu K-edge EXAFS fitting analysis indicates that the local structure (i.e. coordination number and bond distance) of Cu is unaltered before and after the Pd loading (Supplementary Fig. 11 and Supplementary Table 6), which we attribute to the surface loading of Pd (vide infra) and, thus, a minimal change of the Cu local structure was

detected via the bulk-based XAS measurements (Supplementary Fig. 12).

We also examined the XAS at the Pd K-edge and observed that the XANES spectra of the CuPd catalysts resembled that of Pd foil (Fig. 3a), suggesting the metallic character of Pd dopants in the Cu structure. However, a positive shift in the onset of the Pd K-edge XANES of CuPd catalysts compared to that of bulk Pd foil (Supplementary Fig. 13), indicates that the Pd atoms have a depleted electron density due to charge transfer to the neighbouring Cu atoms[28,29]. We also found that the CuPd catalysts exhibit distinct EXAFS oscillations with the Pd foil, which share similar features to Cu at the extended absorption region (Fig. 3a). We confirmed the Cu-analogous EXAFS patterns of $CuPd_{0.007}$ and $CuPd_{0.011}$ at the Pd K-edge when displayed at a radial distance scale (Fig. 3b and Supplementary Fig. 11) via Fourier transform analysis (Methods).

We fit the Pd K-edge EXAFS spectra of different CuPd catalysts and observed a pure Pd–Cu contribution from $CuPd_{0.007}$; whereas, an additional Pd–Pd bond formation was observed for the $CuPd_{0.011}$ due to excess Pd aggregation (Supplementary Fig. 14 and Supplementary Table 5). An overall Pd coordination of ~8 and a Pd–Cu bond length of ~2.58 Å was determined for both CuPd catalysts (Fig. 3c, d and Supplementary Table 6). We note that the low Pd coordination (<12 of Pd foil) extracted from the CuPd catalysts suggests that the Pd dopants preferentially reside at the top surface of the Cu nanoparticles, leading to a ~1.6% bond stretching compared to the Cu–Cu bond (2.54 Å)

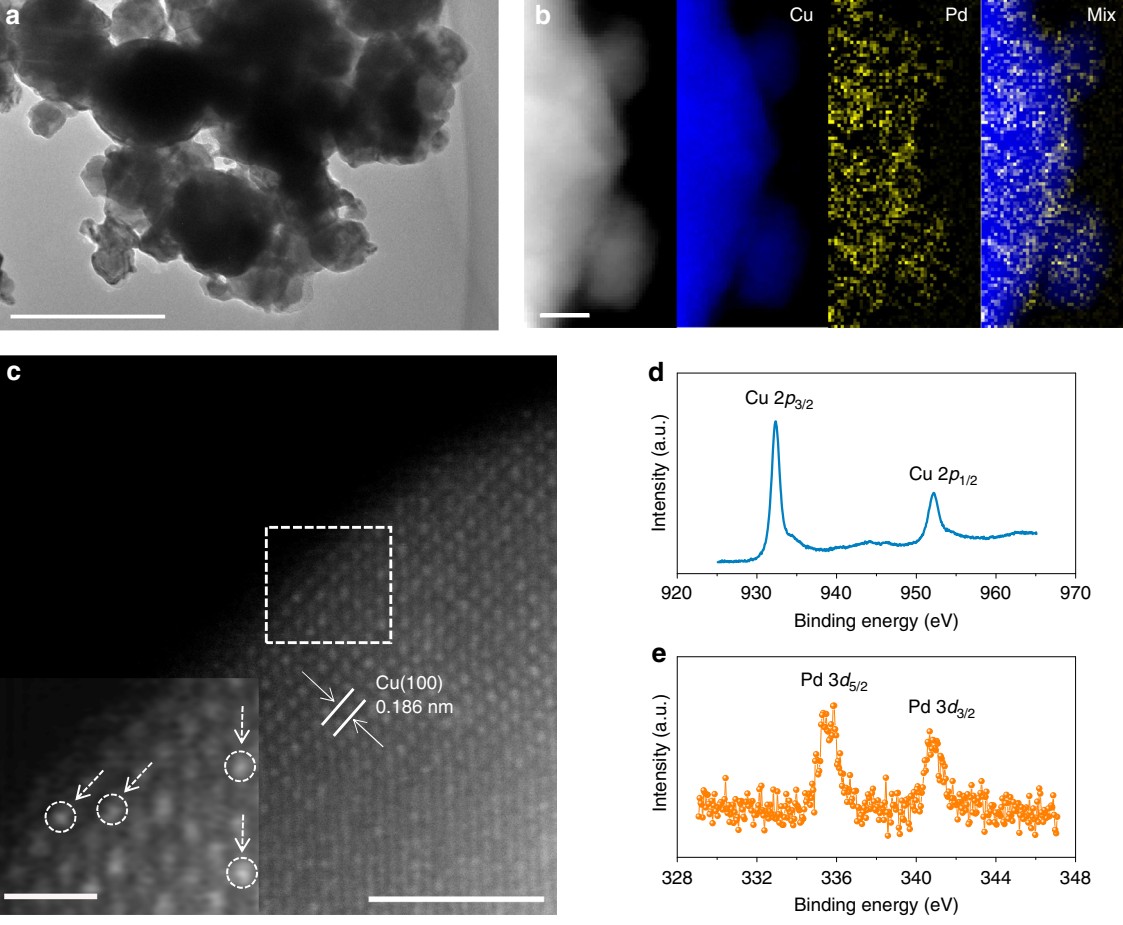

**Fig. 2 Characterization of Pd-doped Cu (CuPd$_{0.007}$) electrocatalysts. a** TEM image of Pd-doped Cu catalysts. Scale bar is 200 nm. **b** High-resolution EELS mapping. Scale bar is 5 nm. **c** HAADF-STEM image and the inverse fast Fourier transition pattern (inset) of selected area. Scale bars are 2 nm for (**c**) and 1/2 nm for the inset, respectively. **d**, **e** X-ray photoelectron spectra of Cu 2p (**d**) and Pd 3d (**e**). All measurements were performed using a Pd-doped Cu electrocatalyst after COR at −0.62 V vs. RHE in 1 M KOH.

simulated from the bulk Cu foil. We also note that the absence of Pd–Pd contribution in CuPd$_{0.007}$ indicates that Pd atoms are atomistically dispersed in the Cu lattice, i.e., Pd is electronically identical to Cu, and the increased Pd loading in CuPd$_{0.011}$ results in the aggregation of excess Pd atoms at the Cu surface, in agreement with previous characterizations[26,29].

**Electrochemical CO reduction performance.** We evaluated the influence of Pd doping of Cu in COR. We deposited the CuPd catalysts with a range of Pd loadings (CuPd$_{0.004}$, CuPd$_{0.007}$ and CuPd$_{0.011}$) on a gas diffusion layer and performed COR measurements in an alkaline flow cell electrolyser[4,18,30]. The experimental loading of Pd in different Cu catalysts determined by ICP-OES is shown in Supplementary Table 5. Selectivities (FE, %) for individual products from COR using different Cu catalysts are provided in Supplementary Fig. 15 and Supplementary Table 7, and the corresponding product activities (current density, mA cm$^{-2}$) are shown in Fig. 4 and Supplementary Fig. 16.

A significant enhancement of both selectivity and activity towards alcohols (ethanol and 1-propanol) at the expense of ethylene was observed (Fig. 4a–c). Operating at a potential window of −0.45~−0.72 V vs. RHE, we found the CuPd$_{0.007}$ catalyst delivered a peak FE$_{alcohol}$ of 40%, accompanied by an alcohol partial current density of 277 mA cm$^{-2}$ (at −0.62 V vs. RHE). This performance represents a 2-fold enhancement of alcohol current density compared to all prior CO/CO$_2$R works at operating current densities >100 mA cm$^{-2}$ (Supplementary

Table 8)[15,17,31–35]. In contrast, the bare-Cu control showed a peak FE$_{alcohol}$ of 27% at −0.66 V vs. RHE, in close agreement with a recent COR report using oxide-derived Cu catalysts[17].

Compared to bare-Cu, trace loadings of Pd (CuPd$_{0.004}$) improve COR selectivity toward alcohol by suppressing ethylene. However, higher Pd loadings (CuPd$_{0.011}$) result in a decrease in FE$_{alcohol}$ and alcohol current density, and a dramatic increase in H$_2$ evolution (Fig. 4d)—a finding consistent with our DFT predictions (Fig. 1c and Supplementary Figs. 4–5) and the presence of Pd aggregates as revealed by the emergence of the Pd–Pd bond in Pd K-edge EXAFS analysis (Fig. 3). The alcohol/ethylene ratio increases sharply from 0.49 on bare-Cu catalysts to 1.07 on CuPd$_{0.007}$ catalysts, and drops to 0.88 on CuPd$_{0.011}$ catalysts due to the aggregation of Pd (Fig. 4e). This volcano-shaped dependence of the alcohol selectivity and activity based on the Pd content suggests that an optimal loading of heteroatoms (Pd) on Cu provides a means to tune H adsorption at the catalyst surface in favour of alcohol production.

To further assess the role of atomic-level vs aggregate form dopants, we prepared a CuPd catalyst with an intermediate Pd:Cu ratio of 0.008 (Supplementary Table 5), similar to the Pd:Cu ratio in CuPd$_{0.007}$, but with Pd aggregation via an alternative galvanic replacement method (Supplementary Fig. 17). We observed a decreased FE$_{alcohol}$ (and increased FE$_{H2}$) in the aggregate case (Supplementary Fig. 18), compared to that of homogeneous Pd dopants (Fig. 4), which further confirms the role of atomic-level doping in steering post-C–C coupling reactions toward alcohols.

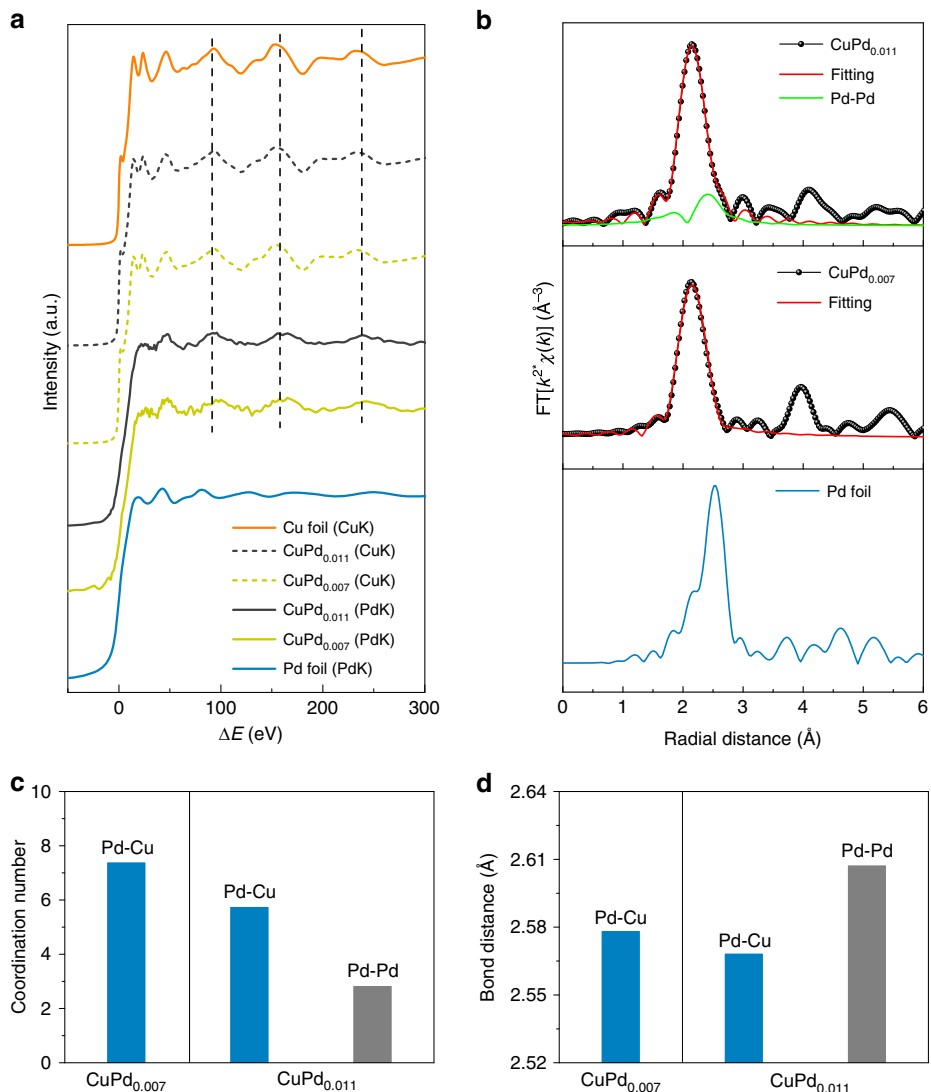

**Fig. 3 X-ray absorption analysis of Pd-doped Cu electrocatalysts. a** A plot of XAFS spectra at a relative energy scale for the Pd K-edge and Cu K-edge of Pd foil, $CuPd_{0.007}$, $CuPd_{0.011}$ and Cu foil by subtracting the Pd K-edge and Cu K-edge photon energies (E) with the absorption thresholds ($E_0$) of Pd 1s (24350 eV) and Cu 1s (8979 eV), respectively, i.e. $\Delta E = E - E_0$. **b** Pd K-edge EXAFS spectra as well as corresponding fitting spectra of Pd foil (bottom), $CuPd_{0.007}$ (middle) and $CuPd_{0.011}$ (top), recorded at a radial distance scale. **c, d** Simulated coordination number (**c**) and bond distance (**d**) of $CuPd_{0.007}$ (left) and $CuPd_{0.011}$ (right) extracted from their Pd K-edge EXAFS. Cu K-edge XAS was performed in operando mode during COR at −0.62 V vs. RHE in 1 M KOH, whereas ex-situ Pd K-edge was conducted after COR at −0.62 V vs. RHE in 1 M KOH.

We monitored the structure and stability of Pd dopants by carrying out a long-term COR test. A polytetrafluorethylene membrane with pore sizes of 0.45 μm was used as gas diffusion layer to avoid catholyte flooding[4]. We successfully demonstrated a stable $FE_{alcohol}$ of 40% over the course of five hours COR operation at a constant current density of 100 mA cm$^{-2}$ (Supplementary Fig. 19). Alcohol selectivity decreased after five hours due to the aggregation of Pd dopants and a partial decrease of Pd loading (Supplementary Fig. 19b, c).

We also prepared a Pt-doped Cu (CuPt) catalyst as Pt was predicted to approach the hydrogenation free energy of Pd in our DFT calculations (Fig. 1). With HAADF-STEM and Pt $L_3$-edge XAS analysis, we confirmed the atomistic electronic structure of the Pt dopants that were uniformly distributed on the surface of Cu nanoparticles (Supplementary Figs. 20, 21). Using a Pt-doped Cu catalyst, we achieved a peak $FE_{alcohol}$ of 36% and a partial current density of 250 mA cm$^{-2}$ at −0.63 V vs. RHE (Supplementary Fig. 22)—levels approaching those achieved with Pd

dopants. Both catalysts demonstrated a twofold enhancement of the $FE_{alcohol}$/$FE_{ethylene}$ ratio compared to bare-Cu catalysts (Supplementary Fig. 23). Electrochemical surface area normalized alcohol current densities further indicate intrinsic alcohol activity (Supplementary Fig. 24). These results surpass prior CO/$CO_2$R reports in selective alcohol activity (Supplementary Table 8).

In summary, we show an efficient metal doping strategy that modulates H adsorption at the catalyst surface to promote hydrogenation of post-C–C coupling reaction intermediates, and thereby boost alcohol formation. Using Pd-doped Cu catalysts we demonstrated a high $FE_{alcohol}$ of 40% with a record alcohol partial current density of 277 mA cm$^{-2}$. This strategy is also effective with Pt dopant, and both CuPd and CuPt catalysts deliver a twofold increase in the alcohol: ethylene production ratio compared to bare copper. We believe this strategy can be further applied to increase efficiencies and selectivities in other catalytic systems involving hydrogenation steps.

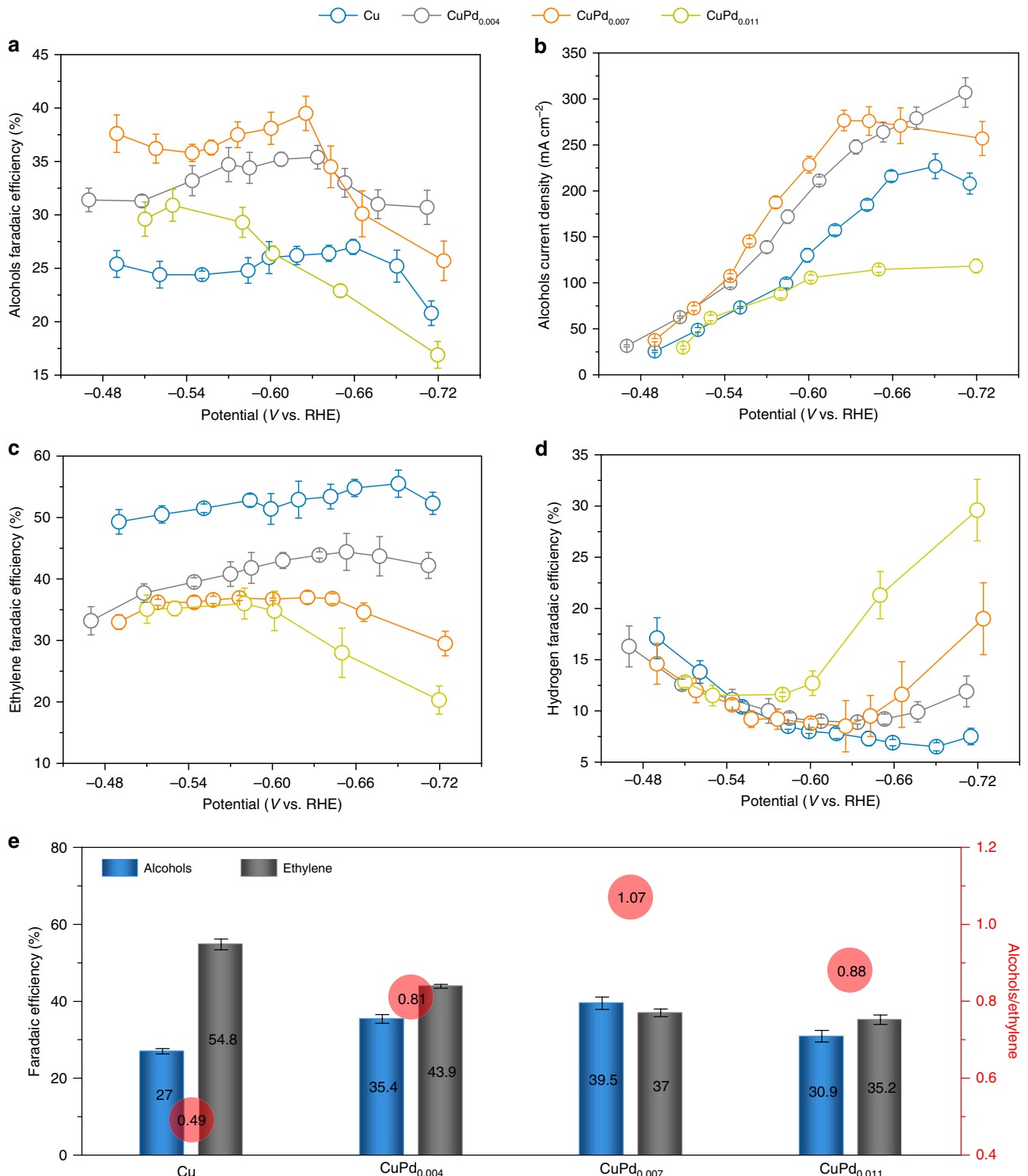

**Fig. 4 COR performance of Pd-doped Cu electrocatalysts. a–d** COR product selectivities (FE, %) and activities (partial current densities, mA cm$^{-2}$) towards alcohols (**a, b**), ethylene (**c**) and hydrogen (**d**) productions on Cu, CuPd$_{0.004}$, CuPd$_{0.007}$ and CuPd$_{0.011}$ catalysts at various applied potentials (vs. RHE) in 1 M KOH. **e** The peak alcohol selectivities and relevant ethylene selectivities on different catalysts at −0.66 V (Cu), −0.63 V (CuPd$_{0.004}$), −0.62 V (CuPd$_{0.007}$) and −0.53 V (CuPd$_{0.011}$) vs. RHE. Numbers in the red circles show the corresponding Faradaic efficiency ratio of alcohol to ethylene. Error bars are means ± SD ($n = 3$ replicates).

## Methods

**DFT calculations**. We performed density functional theory calculations with the Vienna Ab Initio Simulation Package (VASP) code[36,37]. The exchange correlation energy was modelled by using Perdew-Burke-Ernzerhof (PBE) functional within the generalized gradient approximation (GGA)[38]. The projector augmented wave (PAW) pseudo-potentials[39] were used to describe ionic cores. The cut-off energy of 450 eV was adopted after a series of tests. A Methfessel-Paxton smearing of 0.05 eV

to the orbital occupation was applied during the geometry optimization and for the total energy computations. In all calculations, the atoms at all positions have Hellmann–Feynman forces <0.02 eV Å$^{-1}$ and the electronic iterations convergence was 10$^{-5}$ eV using the Normal algorithm. A 6-layer (4 × 4) Cu (111) supercell was built to simulate the exposed surface of copper accompanying with a sufficient vacuum gap of 15 Å. Heteroatoms were exposed on the surface by replacing a Cu atom. Structural optimizations were performed on all modified slab models with a

grid of $(3 \times 3 \times 1)$ k-point. During the adsorption calculations, the top three layers were fully relaxed while the other layers were fixed at the tested lattice positions. The adsorption configurations of intermediates and H are shown in Supplementary Figs. 6 and 7. The vibrational frequencies of free molecules and adsorbates, which are needed to determine zero-point energies and vibrational entropies, were calculated by using the phonon modules. Calculation details for reaction free energies of hydrogenation and dihydroxylation are included in Supplementary Note 1. Surface energy calculations are described in Supplementary Note 2.

**Preparation of electrocatalysts and a GDE**. All reagents in this work were purchased from Sigma Aldrich and used without further purification. Commercial copper nanoparticles (~100 nm particle size, 99.8% trace metals basis) were directly applied as base materials for metal doping, in which we employed a facile and scalable chemical etching method via galvanic replacement under intense ultrasonication[27,40,41]. The energetically favourable galvanic replacement of Cu with Pd/Pt proceeds by Eqs. 3 or 4 below

$$Cu + Pd^{2+} \rightarrow Cu^{2+} + Pd, \qquad (3)$$

$$2Cu + [PtCl_6]^{2-} \rightarrow 2Cu^{2+} + Pt + 6Cl^-. \qquad (4)$$

In the case of adding a small amount of Pd/Pt precursors in the bulk Cu solutions, ultrasound waves consisting of compression and rarefaction cycles not only improve the reaction rate of galvanic replacement but also inhibit the aggregation of Pd/Pt atoms, in agreement with prior works[27,42]. In a typical synthesis process, we first prepared 2 M metal (i.e. Pd and Pt) precursor solutions by diluting PdCl₂ solution (5wt.% in 10 wt.% HCl) or H₂PtCl₆ solution (8 wt.% in H₂O) with deionized (DI) water. Then a volume of 40 mL DI water was added into a flask and deaerated by purging nitrogen for at least 10 min to completely remove air from DI water. Next, 50 mg of copper nanopowder was uniformly dispersed in deaerated DI water using ultrasonication for 15 min. A suitable amount of metal precursor solution (2 M) was then added to initiate selective galvanic replacement under intense ultrasonication for another 15 mins. A continuous N₂ gas flow was applied at all times during synthesis to avoid copper surface oxidization. The resulting doped Cu particles were collected under vacuum filtration using a MF-Millipore membrane filter (0.45 μm pore size), rinsed with deaerated DI water for three times to remove impurities, and dried overnight under vacuum at room temperature (~20 °C). The preparation of a gas diffusion electrode (GDE) involved air-brushing (N₂ as a carrier gas) a catalyst ink consisting of 10 mg of doped Cu nanoparticles, 1 mL isopropanol and 40 μL Nafion solution (~5 wt.%), onto a commercial Sigracet gas diffusion layer (Sigracet 28BC, Fuel Cell Store) with a size of 2 cm × 4 cm. The areal loading amount was ~1 mg cm⁻². After vacuum drying, a $2 \times 2$ cm² of GDE was assembled into a flow cell electrolyser for COR performance.

**Characterization**. XPS measurements were carried out on a K-Alpha XPS spectrometer (PHI 5700 ESCA System), using Al Kα X-ray radiation (1486.6 eV) for excitation. The surface morphology was analyzed using a Hitachi SU9000 SEM/STEM at 2 kV. A Hitachi HF-3300 instrument with an acceleration voltage of 100 kV was employed for TEM analysis. For high-resolution TEM work, an aberration-corrected FEI Titan 80–300 equipped with a CEOS probe and image corrector was used. The microscope was operated at an acceleration voltage of 200 kV and STEM micrographs were captured using a HAADF detector (Fischione). EELS mapping was performed using a Quantum GIF (Gatan) and a K₂ direct electron detector (Gatan). ICP-OES measurements were performed using an iCAP 7400 ICP spectrometer (Thermo Fisher). P-XRD was performed on a MiniFlex600 instrument with a copper target ($\lambda = 1.54056$ Å) at room temperature. Operando XAS measurements at the Cu K-edge and Pt L₃-edge were carried out at the 9BM beamline of Advanced Photon Source (APS, Argonne National Laboratory, Lemont, Illinois) in a flow cell[16]. Ex-situ Pd K-edge experiments were conducted at the 20BM beamline of APS by sealing samples in Kapton tape right after the COR testing. Fluorescence yield was recorded using silicon drift detectors at both the 20BM and 9BM beamlines.

**EXAFS analysis**. A standardized IFEFFIT package (including Athena and Artemis softwares) was employed to analyze the EXAFS data[43]. We first conducted energy calibration (with metal foil standards) and spectral normalization using Athena. Then we performed a spectral transformation from an E scale (photon energy), to a k scale (photoelectron wave vector), and finally to the R scale (radial distance) using Fourier transformation (k²-weighted). Afterward, EXAFS data recorded in k scale was exported from Athena and imported in Artemis for fitting analysis. The fittings of the first shells of Cu, Pd and Pt elements were carried out in FEFF[44], in which structural parameters of absorbing atoms were calculated including coordination number (CN), bond distance (R), inner potential shift ($\Delta E_0$) and Debye-Waller factor ($\sigma^2$).

**Electrochemical reduction of carbon monoxide**. All COR experiments were performed using a three-electrode flow cell electrolyser[4,16]. As-prepared GDE (details above), Ag/AgCl (filled with 1 M KCl) and nickel foam (1.6 mm thickness, MTI Corporation) were used as cathode, reference electrode and anode, respectively. 1 M KOH solutions were used as electrolytes at both cathode and anode sides. An anion

exchange membrane (Fumasep FAA-PK-130) was used to separate the catholyte and anolyte compartments and hence avoid product crossover. An Autolab PGSTAT204 (Metrohm Autolab) in combination with a BOOSTER10A module (Metrohm Autolab) was used as a power supply. During COR experiments, aqueous KOH solution was directed into the cathode and anode compartments via dedicated variable-speed peristaltic tubing pumps (Control Company 3385). A continuous flow of CO at a rate of 30 s.c.c.m. was directed into the gas compartment and reacted at the catalyst-catholyte interface via gas diffusion through the GDL.

Potentials reported in this work were calculated to the RHE reference scale using Eq. 5 below

$$E_{RHE} = (E_{Ag/AgCl})_{iR} + 0.235 \text{ V} + 0.059 \times \text{pH}. \qquad (5)$$

The ohmic drop correction (i.e. iR compensation: i is the applied current and R is the cell resistance) was conducted using Eq. 6 below

$$(E_{Ag/AgCl})_{iR} = E_{Ag/AgCl} - 0.85 \times i \times \text{R}, \qquad (6)$$

where $E_{Ag/AgCl}$ was the applied potential before iR compensation and a R value of 3.4 Ω was determined by performing an electrochemical impedance spectroscopy measurement using an Autolab PGSTAT302N electrochemical workstation coupled with a FRA32M module. A factor of 0.85 is applied in iR compensation during flow cell operation due to a low resistivity of 1 M KOH electrolyte which holds a relatively low voltage drop over the electrolyte[18].

Gas products were analyzed using a gas chromatography (GC, PerkinElmer Clarus 680) equipped with a Molecular Sieve 5A capillary column and a packed Carboxen-1000 column. Argon (Linde, 99.999%) was used as a carrier gas. Hydrogen and ethylene products were quantified using a thermal conductivity detector and a flame ionization detector, respectively, equipped with a methanizer loaded in GC instrument (Supplementary Fig. 25). Liquid products dissolved in catholyte were quantified using one-dimensional ¹H Nuclear magnetic resonance spectroscopy (¹H NMR) coupled with a Agilent DD2 500 spectrometer, in which a diluted Dimethyl sulfoxide (DMSO) in D₂O was used as an internal standard for the identification and quantification of liquid products (Supplementary Fig. 26).

## Data availability
The data that support the findings of this study are available from the corresponding author on reasonable request.

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

## Acknowledgements

This work was supported financially by the Ontario Research Fund Research-Excellence Program, the Natural Sciences and Engineering Research Council (NSERC) of Canada, the CIFAR Bio-Inspired Solar Energy program, and the University of Toronto Connaught grant. This research used synchrotron resources of the Advanced Photon Source (APS), an Office of Science User Facility operated for the U.S. Department of Energy (DOE) Office of Science by Argonne National Laboratory, and was supported by the U.S. DOE under Contract No. DE-AC02-06CH11357, and the Canadian Light Source and its funding partners. The authors thank Prof. M. Graetzel at the École polytechnique fédérale de Lausanne for insightful discussions, Dr. T.P. Wu, Dr. Y.Z. Finfrock and Dr. L. Ma for technical support at 9BM and 20BM beamlines of APS. D.S. acknowledges the NSERC E.W.R Steacie Memorial Fellowship. J.L. acknowledges the Banting Postdoctoral Fellowships program. Lab infrastructure funding from the Canada Foundation for Innovation and the Ontario Research Fund is gratefully acknowledged. We acknowledge the Toronto Nanofabrication Centre (TNFC) and the Ontario Centre for the Characterization of Advanced Materials (OCCAM) for sample preparation and characterization facilities. All DFT computations were performed on the Niagara supercomputer at the SciNet HPC Consortium. SciNet is funded by: the Canada Foundation for Innovation; the Government of Ontario; Ontario Research Fund Research Excellence Program; and the University of Toronto.

## Author contributions

E.H.S. and D.S. supervised the project. J.L. conceived the idea and carried out all the experiments. A.N.X. performed and Z.Y.W. assisted the DFT simulations. J.L., F.W.L., D. H.N., Y.W.L., Z.Q.W., J.T.C. and T.K.S. performed and analyzed synchrotron data. C.Q. Z., B.C., X.W.D. and B.Z. helped SEM and TEM analysis. J.W. conducted XPS measurements. Y.Z.W. carried out ICP-OES tests. C.M.G., Y.H.W., A.O., Y.X., T.T.Z. and M. C.L. assisted electrochemical experiments and analysis. All authors discussed the results and assisted during manuscript preparation.

## Competing interests

The authors declare no competing interests.
