## [Peer Review File · Nature Communications]

Reviewers' Comments:

Reviewer #1:

Remarks to the Author:

The work by Li and Xu et. al. presents a system with a record selectivity for the production of oxygenate species through carbon monoxide reduction using a Pd-doped Cu surface. The experiments are well thought out and well realised and lead to a system whose activity warrants publication in Nature Communications. Nevertheless, the theoretical interpretation of the data has a few inconsistencies to be rectified before publication.

1. While it is clear that addition of Pd and Pt increase the selectivity for oxygenates, the emphasis on single-atom Pd sites being key to the high selectivity is not sufficiently proven for oxygenates over ethylene. In Figure 4e it appears that the aggregated Pd catalyst still shows high selectivity for oxygenate vs. ethylene formation, but the aggregates have reduced overall selectivity for CO reduction due to competing proton reduction, which is expected given the propensity of Pd for H₂ evolution.

To prove that aggregates are worse at oxygenate vs. ethylene selectivity, an experiment comparing CuPd_{0.015} with single atom and aggregated Pd is shown, however it is not clear if these two electrodes have the same Pd content. How was Pd content of the aggregated CuPd_{0.015} analysed? This must be added to the text. If it is the same galvanic exchange solution (as used for the single atom CuPd_{0.015}) but with a different preparation procedure (as is implied by the heading of Figure S15), it should not be assumed that the electrode content is still CuPd_{0.015}. This comment stems from the fact that to create an electrode containing the exact same amount of Pd but in a different morphology would be difficult without optimisation.

That is not to say the data are not very interesting; doping with a small number of hydrogenation sites to increase oxygenate selectivity is an important finding. I recommend the authors concentrate on the use of single-atom Pd sites for creating a surface with high oxygenate/low H₂ selectivity. The use of single atoms vs. aggregated catalysts for superior oxygenate formation over ethylene is only proven in the DFT data.

2. Can the authors provide the Cu:Pd ratio through a bulk technique to compliment their characterisation? For example, Inductively coupled plasma atomic emission spectroscopy

3. It is unusual to plot 'peak selectivity' across a range of surfaces without indication of the potential at which the 'peak' was achieved, as in Figure 4e. It would also be useful to see error bars on these graphs.

4. Why is pure Cu not taken to the most negative potentials? It would be useful to compare to the activity of the Pd-doped Cu at high overpotential.

5. The authors should show the Faradaic efficiency for every product for each electrode composition at each potential, rather than just the 'oxygenates' vs. 'ethylene' produced. This should be at least shown in a Supplementary Table as this information would be necessary for other researchers to compare their own selectivity towards certain products, such as ethanol and propanol.

6. The authors should specify the nature of the GDL more specifically in the methods, as 'Sigracet' GDLs are available in a variety of compositions.

7. Indication of the ohmic drop correction (if any) should be described in the methodology.

8. Line 187 – oxygenate is misspelled

9. Line 262 – area is misspelled

10. Figure S23b – roughness is misspelled

If these comments are addressed I can recommend this work be published as soon as possible.

Reviewer #2:

Remarks to the Author:

In this manuscript, Li et al. reported a novel strategy, modulating hydrogen adsorption, to tune the selectivity of oxygenates in CO reduction and achieved a remarkable efficiency for CO-to-oxygenates conversion. Benefiting from the configuration of gas diffusion electrode, the partial current density reached $> 300 \text{ mA cm}^{-2}$. The selectivity and partial current density of oxygenates are certainly beyond the state-of-the-art in CO conversion. While I have my reservations regarding the DFT calculations, the performance of the catalyst and reputation of the group in this field make me believe that this work is worthy of publication in Nature Communications after the following comments are properly addressed.

1. The proposed mechanism for the production of oxygenates, as well as the entire manuscript, is largely based on the result from Goddard and co-workers (ref. 20-22). However, the electrocatalysis community have not reached a consent regarding this mechanism since many other researchers believed the CO dimer mechanism (Angew Chem. Int. Ed. 52, 7282-7285). Moreover, the calculations in Goddard's papers are not fully consistent and may require additional caution (see the first few lines of JACS 141, 4191-4193). It is more plausible that increased hydrogen adsorption will facilitate the formation of ethylene instead of ethanol since the pathway towards ethylene is likely to require more hydrogen than the formation of ethanol (Nat. Energy 4, 732-745). It seems that the authors have also agreed on the CO dimer mechanism in their previous publications (Nat. Catal. 1, 421-428) and this inconsistency shall be clarified. The backbone of the manuscript, H adsorption facilitating oxygenates formation, seems to be too weak.
2. While single atom Pd catalyst is used in DFT, the authors do not successfully synthesize single atom Pd to modify Cu. At least, the evidence of single atom Pd is not very strong since there is a lack of atomic resolution STEM. This is very misleading and may confuse the readers. Nanocluster, instead of single atom, is a more appropriate term and should be used in DFT calculations.
3. XPS is used to determine the surface ratio of Cu to Pd and the authors admitted that this method may underestimate the amount of Pd due to the probing depth of XPS. Since this information is very important for the readers and those who want to reproduce the results, have the authors considered electrochemical methods such as Pb UPD, which is more surface-sensitive, to determine the amount of Pd?
4. A detailed breakdown of the oxygenates is necessary in Figure 4 since 'total FE of oxygenates' is a tricky term. It is well known that CO reduction process generates a lot of acetate, which cannot be directly used as fuel additive (only ethanol and n-propanol can be directly used). More importantly, the mechanism of acetate formation is likely to be significantly different from the formation of ethanol/n-propanol and simply adding them together is not acceptable (JACS 141, 4191-4193). The comparison with literature shall also be based on the generation of alcohols instead of oxygenates. The ratio between oxygenates/ethylene shall also be changed to alcohols/ethylene, as shown in the authors' previous publication, Nat. Catal. 1, 421-428.
5. It is well known that nanoclusters may undergo agglomeration after long term electrolysis. The performance of the catalysts seemed to be evaluated over a time period of only 40 min. The stability of catalysts over >10 hours catalysis, as well as the change of catalysts, shall be presented. The stability is a critical parameter for catalyst and shall be discussed in detail.
6. The review of previous reports is out of date and shall be updated. For example, the authors quoted 'even CO₂-CO-oxygenates tandem catalysts fail to exceed 40% oxygenate selectivity'. However, even the author's own group has recently reported CuAg catalyst that showed 41% ethanol selectivity (JACS 141, 8584-8591) and many of similar reports exist.

Reviewer #3:

Remarks to the Author:

Sinton, Sargent and coworkers have produced a highly interesting paper on the reduction of CO to oxygenates. Guided by DFT calculations, they find that predict that isolated Cu atoms on Pd should enhance oxygenate selectivity. They perform experiments, that putatively verify this claim.

Although this paper is highly interesting to the broad readership of Nature Communications, it has some ambiguities, which, to my mind, prevent its publication in the current form. More specifically:

- 1) It is unclear what products are formed. The data in their present form lack transparency. To this end:
 - a. The authors should show the raw NMR/GC data for both their calibration and the actual reaction products, for each and every product. They should plot in their main text the breakdown for each product at every potential, rather than bunching all oxygenates together.
 - b. They should plot the partial current densities for each and every product at each potential.

- c. They should also plot the unbalanced charge for at each potential
- 2) The authors should calibrate the reference electrode experimentally using a real RHE.
 - 3) They should also correct for Ohmic losses and state exactly how they measured Ohmic losses.
 - 4) The authors should provide more convincing evidence to explain why their synthesis method should favour single atom catalysts.
 - 5) All data are performed ex-situ, meaning that the surface atoms will inevitably be oxidised by air transfer and hence unrepresentative of the catalytically active state. Soriaga and coworkers have demonstrated how mobile Cu atoms are under CO₂ reduction conditions. For instance, reference 9 shows that a catalyst that forms single atom Cu reversibly aggregates under reaction conditions and then reverts to the single atom form ex-situ. To this end, it would be far more convincing if the authors would provide in-situ spectroscopic data to demonstrate that they have single atom catalysts, either based on FTIR (see works by Peter Stair or Maria Stephanopoulos for gas phase reactions) or XAS (reference 9 of current text).

If the authors are able to convincingly address my concerns above, then their work might be suitable for publication.

Manuscript ID: NCOMMS-19-2396360

“Enhanced multi-carbon alcohol electroproduction from CO via modulated hydrogen adsorption”

Reviewer 1

The work by Li and Xu et. al. presents a system with a record selectivity for the production of oxygenate species through carbon monoxide reduction using a Pd-doped Cu surface. The experiments are well thought out and well realised and lead to a system whose activity warrants publication in Nature Communications. Nevertheless, the theoretical interpretation of the data has a few inconsistencies to be rectified before publication.

We thank Review #1 for this feedback.

1. While it is clear that addition of Pd and Pt increase the selectivity for oxygenates, the emphasis on single-atom Pd sites being key to the high selectivity is not sufficiently proven for oxygenates over ethylene. In Figure 4e it appears that the aggregated Pd catalyst still shows high selectivity for oxygenate vs. ethylene formation, but the aggregates have reduced overall selectivity for CO reduction due to competing proton reduction, which is expected given the propensity of Pd for H₂ evolution.

To prove that aggregates are worse at oxygenate vs. ethylene selectivity, an experiment comparing CuPd_{0.015} with single atom and aggregated Pd is shown, however it is not clear if these two electrodes have the same Pd content. How was Pd content of the aggregated CuPd_{0.015} analysed? This must be added to the text. If it is the same galvanic exchange solution (as used for the single atom CuPd_{0.015}) but with a different preparation procedure (as is implied by the heading of Figure S15), it should not be assumed that the electrode content is still CuPd_{0.015}. This comment stems from the fact that to create an electrode containing the exact same amount of Pd but in a different morphology would be difficult without optimisation.

That is not to say the data are not very interesting; doping with a small number of hydrogenation sites to increase oxygenate selectivity is an important finding. I recommend the authors concentrate on the use of single-atom Pd sites for creating a surface with high oxygenate/low H₂ selectivity. The use of single atoms vs. aggregated catalysts for superior oxygenate formation over ethylene is only proven in the DFT data.

We performed inductively coupled plasma optical emission spectrometry (ICP-OES) for all CuPd samples as suggested by the reviewer. We also renamed all samples according to *the Cu:Pd ratios applied in the precursor solutions* which correlate well with the Cu:Pd ratios determined by ICP-OES shown in **Supplementary Table 5** (i.e. CuPd_{0.010}, CuPd_{0.015} and CuPd_{0.022} have now renamed as CuPd_{0.005}, CuPd_{0.010} and CuPd_{0.015}, respectively).

The Pd contents of CuPd_{0.015} catalysts (now termed CuPd_{0.010} since a Pd:Cu ratio of 0.010 in the precursor solution was used during synthesis) in both atomic and aggregate forms were determined by inductively coupled plasma optical emission spectrometry (ICP-OES). A similar Cu:Pd bulk ratio was found in both CuPd_{0.010} samples.

Supplementary Table 5. Elemental composition of Cu and Pd in different CuPd catalysts determined by inductively coupled plasma optical emission spectrometry (ICP-OES).

Sample ID	Cu ($\mu\text{g mL}^{-1}$)	Pd ($\mu\text{g mL}^{-1}$)	Cu:Pd ratio
CuPd _{0.005}	563.1	3.4	1:0.004
CuPd _{0.010}	473.9	5.6	1:0.007
CuPd _{0.010} (aggregate)	885.4	12.4	1:0.008
CuPd _{0.015}	464.7	8.8	1:0.011

We added the following to the revised manuscript (**Page 9, Line 191**):

“A similar Cu:Pd ratio was determined by ICP-OES (**Supplementary Table 5**), suggesting the same Pd loading at the Cu surface in both CuPd_{0.010} catalysts.”

2. Can the authors provide the Cu:Pd ratio through a bulk technique to compliment their characterisation? For example, Inductively coupled plasma atomic emission spectroscopy

We performed inductively coupled plasma optical emission spectrometry (ICP-OES) to determine the bulk ratio of Cu:Pd in all of our catalysts. The results are tabulated above and in the revised SI (**Supplementary Table 5**).

We further clarify this aspect in the text:

Page 7, Line 128: “Using a Pd:Cu ratio of 0.01 in the precursor solution, a similar Pd:Cu ratio of 0.007 was achieved, as measured by inductively coupled plasma atomic emission spectroscopy (ICP-OES, **Supplementary Table 5**).”

Page 8, Line 164: “We deposited the CuPd catalysts with a range of Pd loadings (CuPd_{0.005}, CuPd_{0.010} and CuPd_{0.015}) on a gas diffusion layer and performed COR measurements in an alkaline flow cell electrolyser^{4, 18, 30}. The experimental loading of Pd in different Cu catalysts determined by ICP-OES is shown in **Supplementary Table 5**.”

3. It is unusual to plot ‘peak selectivity’ across a range of surfaces without indication of the potential at which the ‘peak’ was achieved, as in Figure 4e. It would also be useful to see error bars on these graphs.

Potentials at the peak FE_{alcohols} for different Cu catalysts have been added in the Figure caption of **Fig. 4e**. Error bars are also included.

Page 24, Line 488: “The peak alcohol selectivities and relevant ethylene selectivities on different catalysts at -0.66 V (Cu), -0.63 V (CuPd_{0.005}), -0.62 V (CuPd_{0.010}) and -0.53 V (CuPd_{0.015}) vs. RHE. Numbers in the red circles show the corresponding Faradaic efficiency ratio of alcohol to ethylene. Error bars are means \pm SD (n = 3 replicates).”

4. Why is pure Cu not taken to the most negative potentials? It would be useful to compare to the activity of the Pd-doped Cu at high overpotential.

We now extend the COR performance of pure Cu to large negative potentials in our revised Fig. 4.

Fig. 4. COR performance of Pd-doped Cu electrocatalysts. a-d, COR product selectivities (FE, %) and activities (partial current densities, mA cm⁻²) towards alcohols (a, b), ethylene (c) and hydrogen (d) productions on Cu, CuPd_{0.005}, CuPd_{0.010} and CuPd_{0.015} catalysts at various applied potentials (vs. RHE) in 1 M KOH. **e,** The peak alcohol selectivities and relevant ethylene selectivities on different catalysts at -0.66 V (Cu), -0.63 V (CuPd_{0.005}), -0.62 V (CuPd_{0.010}) and -0.53 V (CuPd_{0.015}) vs. RHE. Numbers in the red circles show the corresponding Faradaic efficiency ratio of alcohol to ethylene. Error bars are means ± SD (n = 3 replicates).

5. The authors should show the Faradaic efficiency for every product for each electrode composition at each potential, rather than just the 'oxygenates' vs. 'ethylene' produced. This should be at least shown in a Supplementary Table as this information would be necessary for other researchers to compare their own selectivity towards certain products, such as ethanol and propanol.

We updated **Fig. 4** shown above. Detailed product distributions of different Cu catalysts from COR are given in **Supplementary Table 7**.

Supplementary Table 7. A summary of Faradaic efficiencies for all products at different Cu catalysts.

Catalyst	Potential (V vs. RHE)	Faradaic Efficiency (%)					
		Hydrogen	Ethylene	Acetate	Ethanol	1-Propanol	Total
Cu	-0.49	17.1±2	49.3±2	5.9±0.5	11.6±2	13.8±0.5	97.7
	-0.52	13.8±1.1	50.5±1.4	5.6±0.3	13.7±1.5	10.7±1	94.3
	-0.55	10.4±0.5	51.5±0.7	6.1±0.2	14.3±0.5	10.1±0.2	92.4
	-0.58	8.5±0.3	52.8±1	8.7±1.2	15.8±1.2	9±1.2	94.8
	-0.60	8±0.2	51.4±2.5	8.1±2	17.4±1	8.6±2	93.5
	-0.62	7.8±0.5	52.9±3	9.2±1.5	17.8±1.2	8.4±0.5	96.1
	-0.64	7.3±0.7	53.4±2	9.8±1	18.6±1.2	7.8±0.3	96.9
	-0.66	6.9±0.3	54.8±1.4	9±1.3	20.1±0.8	6.9±0.6	97.7
	-0.69	6.5±0.4	55.5±2.2	11.2±2.1	19.8±2.5	5.4±0.5	98.4
	-0.71	7.5±0.8	52.3±1.8	14.4±1.8	17.7±2	3.1±0.3	95
CuPd _{0.005}	-0.47	16.3±2	33.2±2.3	8.1±0.5	17.8±1.2	13.6±1	89
	-0.51	12.6±0.5	37.7±1.5	9.2±1.5	21.8±0.6	9.5±0.5	90.8
	-0.54	11.1±1	39.5±0.7	11.5±2	23.5±1.1	9.7±0.7	95.3
	-0.57	10±1.2	40.8±2	11.5±1.2	25.9±2	8.8±0.2	97
	-0.59	9.3±0.5	41.8±2.5	9.1±1	26.3±1.4	8.1±0.5	94.6
	-0.61	9±0.3	43±1.3	10.6±0.7	28±1	7.2±0.3	97.8
	-0.63	8.9±0.2	43.9±0.5	11.6±1.3	28.5±1.2	6.9±1	99.8
	-0.65	9.2±0.5	44.4±3	12.7±1	27.4±0.5	5.6±0.4	99.3
	-0.68	9.9±1	43.7±3.2	11.4±1.4	26.2±1.2	4.8±0.2	96
	-0.71	11.9±1.5	42.2±2.1	11.4±1.5	26.1±1	4.6±0.2	96.2
CuPd _{0.010}	-0.49	14.6±2	33±1.2	7.7±1.2	23.3±1.5	14.3±2	92.9
	-0.52	12±1.2	36.2±0.5	11.3±2	25.9±0.7	10.3±1.5	95.7
	-0.54	10.6±0.5	36.2±1	12.9±0.5	27.1±1.3	8.7±0.3	95.5
	-0.56	9.2±0.8	36.6±0.6	13.1±1.3	28.7±1.2	7.6±0.2	95.2
	-0.58	9.2±1	36.9±0.4	13.9±1.1	30.1±2.1	7.4±0.3	97.5
	-0.60	8.8±0.4	36.7±0.2	14.5±0.4	31.5±0.8	6.6±0.5	98.1
	-0.62	8.5±2.5	37±1	14.9±1.2	33.2±1	6.3±0.2	99.9
	-0.64	9.5±2	36.8±0.8	18.6±1.3	29.2±1.2	5.3±0.7	99.4
	-0.67	11.6±3.2	34.6±1.5	19.2±0.7	26.1±2	4±0.3	95.5
	-0.72	19±3.5	29.5±2	21±2.1	22.4±1.5	3.3±0.2	95.2
CuPd _{0.015}	-0.51	12.8±0.5	35.1±2.3	5.7±0.5	21.2±1.2	8.4±1	83.2
	-0.53	11.5±1	35.2±1.2	8.2±1.2	25.8±1.5	5.1±0.5	85.8
	-0.58	11.6±0.2	36±2.5	11.7±2.3	25.1±1.5	4.2±1.3	88.6

	-0.60	12.7±1.2	34.8±3.2	14.4±1.4	23.7±1	2.7±0.4	88.3
	-0.65	21.3±2.3	28±4	19.2±1.2	21.4±0.7	1.5±0.5	91.4
	-0.72	29.6±3	20.3±2.3	19.5±2	16.3±2.3	0.6±0.2	86.3

We added the following in the revised manuscript (**Page 8, Line 167**):

“Selectivities (FE, %) for individual products from COR using different Cu catalysts are provided in **Supplementary Fig. 15** and **Supplementary Table 7**, and the corresponding product activities (current density, mA cm⁻²) are shown in **Fig. 4** and **Supplementary Fig. 16**.”

6. The authors should specify the nature of the GDL more specifically in the methods, as ‘Sigracet’ GDLs are available in a variety of compositions.

Page 13, Line 268: The *Methods* section is now clear (Sigracet 28BC, Fuel Cell Store).

7. Indication of the ohmic drop correction (if any) should be described in the methodology.

We have added the ohmic drop correction details in the revised manuscript (**Page 14, Line 308**):

“Potentials reported in this work were calculated to the reversible hydrogen electrode (RHE) reference scale using $E_{\text{RHE}} = (E_{\text{Ag/AgCl}})_{iR} + 0.235 \text{ V} + 0.059 \times \text{pH}$. The ohmic drop correction (i.e. iR compensation: i is the applied current and R is the cell resistance) was conducted using $(E_{\text{Ag/AgCl}})_{iR} = E_{\text{Ag/AgCl}} - 0.85 \times i \times R$, in which $E_{\text{Ag/AgCl}}$ was the applied potential before iR compensation and a R value of 3.4Ω was determined by performing an electrochemical impedance spectroscopy measurement using an Autolab PGSTAT302N electrochemical workstation coupled with a FRA32M module. A factor of 0.85 is applied in iR compensation during flow cell operation due to a low resistivity of 1 M KOH electrolyte which holds a relatively low voltage drop over the electrolyte.¹⁸”

8. Line 187 – oxygenate is misspelled

9. Line 262 – area is misspelled

10. Figure S23b – roughness is misspelled

All corrections have been made to the revised files.

If these comments are addressed I can recommend this work be published as soon as possible.

We thank Reviewer #1 for these helpful comments.

Reviewer #2

In this manuscript, Li et al. reported a novel strategy, modulating hydrogen adsorption, to tune the selectivity of oxygenates in CO reduction and achieved a remarkable efficiency for CO-to-oxygenates conversion. Benefiting from the configuration of gas diffusion electrode, the partial current density reached > 300 mA cm⁻². The selectivity and partial current density of oxygenates are certainly beyond the state-of-the-art in CO conversion. While I have my reservations regarding the DFT calculations, the performance of the catalyst and reputation of the group in this field make me believe that this work is worthy of publication in Nature Communications after the following comments are properly addressed.

We thank Reviewer #2 and detail below our responses that seek to address the reviewer's comments.

1. The proposed mechanism for the production of oxygenates, as well as the entire manuscript, is largely based on the result from Goddard and co-workers (ref. 20-22). However, the electrocatalysis community have not reached a consent regarding this mechanism since many other researchers believed the CO dimer mechanism (Angew Chem. Int. Ed. 52, 7282-7285). Moreover, the calculations in Goddard's papers are not fully consistent and may require additional caution (see the first few lines of JACS 141, 4191-4193). It is more plausible that increased hydrogen adsorption will facilitate the formation of ethylene instead of ethanol since the pathway towards ethylene is likely to require more hydrogen than the formation of ethanol (Nat. Energy 4, 732-745). It seems that the authors have also agreed on the CO dimer mechanism in their previous publications (Nat. Catal. 1, 421-428) and this inconsistency shall be clarified. The backbone of the manuscript, H adsorption facilitating oxygenates formation, seems to be too weak.

We clarify in the revision that the mechanism we proposed, hydrogenation vs. dehydroxylation, does not conflict with the CO dimer mechanism. In this work we focus on tuning the selectivity of alcohol vs. hydrocarbon, with the focus on steering the post-C-C coupling reaction steps.

We removed the paper (*J. Am. Chem. Soc.* **140**, 9337–9340 (2018)) and instead cite the mechanism proposed in Ref 19 (*J. Am. Chem. Soc.* **139**, 130-136 (2017)) and Ref 20 (*Proc. Natl. Acad. Sci. U. S. A.* **114**, 1795–1800 (2017)), which is similar to the mechanism proposed by Koper et al. (*Nat. Energy* **4**, 732-745 (2019) and *Angew Chem. Int. Ed.* **52**, 7282-7285 (2013)), (now cited as Ref 2 and Ref 22).

In particular, Ref 2 (*Nat. Energy* **4**, 732-745 (2019)) indicates "... these C₂ species are formed through common intermediates... the hydrogenation of which leads to acetaldehyde and subsequently ethanol, and the hydrogenolysis of which leads to ethylene", which aligns with our proposed mechanism for favouring alcohols over ethylene, i.e. hydrogenation for alcohol vs. hydrogenolysis (dehydroxylation) for ethylene.

We note that our recent work in promoting ethanol with hydroxides (Ref 31: *Nat. Commun.* **10**, 5814 (2019)) further supports the hypothesis proposed in this work, showing that hydrogenation of C₂ intermediate is key to promoting ethanol generation.

We also note that the production of ethylene and ethanol from CO require the same hydrogen stoichiometry:

We clarified these aspects in the revised manuscript (**Page 4, Line 70**):

“CO dimerization has been suggested as the rate-determining step for CO-to-C₂+ conversion, generally^{2, 21, 22}. Recent works by Goddard and co-workers^{19, 20} have shown that the reaction of the intermediate HOCCH* through either a hydrogenation pathway (i.e. $\text{HOCCH}^* + \text{H}^* \rightarrow \text{CHCHOH}^*$) or a dehydroxylation pathway (i.e. $\text{HOCCH}^* + \text{e}^- \rightarrow \text{CCH}^* + \text{OH}^-$) (**Fig. 1a** and **Supplementary Fig. 1**) determines the selectivity towards alcohol vs. ethylene, which is similar to the mechanism proposed by Koper and co-workers^{2, 22}, showing that hydrogenation of C₂ intermediates leads to acetaldehyde and subsequently ethanol generation. We reasoned that controlling the catalytic hydrogenation of HOCCH* intermediates could steer high-rate COR selectivity from ethylene to alcohols.”

2. While single atom Pd catalyst is used in DFT, the authors do not successfully synthesize single atom Pd to modify Cu. At least, the evidence of single atom Pd is not very strong since there is a lack of atomic resolution STEM. This is very misleading and may confuse the readers. Nanocluster, instead of single atom, is a more appropriate term and should be used in DFT calculations.

We performed atomic resolution HAADF-STEM and included the results in revised **Fig. 2**. Individual Pd atoms are now resolved on the Cu crystal surface via the difference in Z-contrast.

Fig. 2. Characterization of Pd-doped Cu electrocatalysts. **a**, TEM image of Pd-doped Cu catalysts. **b**, High-resolution EELS mapping. **c**, HAADF-STEM image and the inverse fast Fourier transition pattern (inset) of selected area. **d**, **e**, X-ray photoelectron spectra of Cu 2p (**d**) and Pd 3d (**e**). All measurements were performed using a Pd-doped Cu electrocatalyst after COR at -0.62V vs. RHE in 1 M KOH.

We now include these in the revised manuscript (**Page 6, Line 121**):

“The Pd dopants were determined to be evenly distributed in the Cu structure using the aberration-correction high-angle annular dark-field scanning transmission electron microscopy (HAADF-STEM), coupled with electron energy loss spectroscopy (EELS) mapping (**Fig. 2b** and **2c**), as well as energy-dispersive X-ray (EDX) mapping (**Supplementary Fig. 10**). Individual Pd atoms were discerned on the crystal surface of Cu in the Pd-doped Cu catalysts based on the difference in Z-contrast.”

3. XPS is used to determine the surface ratio of Cu to Pd and the authors admitted that this method may underestimate the amount of Pd due to the probing depth of XPS. Since this information is very important for the readers and those who want to reproduce the results, have the authors considered electrochemical methods such as Pb UPD, which is more surface-sensitive, to determine the amount of Pd?

We have performed Pb UPD of the CuPd_{0.010} catalysts and found that no Pb deposition on Pd is observed, consistent with literature (*J. Electrochem. Soc.* **165**, J3074-J3082 (2018)).

Figure R1. A CV curve of Pb UPD of CuPd_{0.010}.

To further support these measurements, we quantified the Cu:Pd bulk ratio of various Cu catalysts using inductively coupled plasma optical emission spectrometry. The resulting Cu:Pd ratios are tabulated below and included in the revised SI.

Supplementary Table 6. Elemental composition of Cu and Pd in different CuPd catalysts determined by inductively coupled plasma optical emission spectrometry (ICP-OES).

Sample ID	Cu ($\mu\text{g mL}^{-1}$)	Pd ($\mu\text{g mL}^{-1}$)	Cu:Pd ratio
CuPd _{0.005}	563.1	3.4	1:0.004
CuPd _{0.010}	473.9	5.6	1:0.007
CuPd _{0.010} (aggregate)	885.4	12.4	1:0.008
CuPd _{0.015}	464.7	8.8	1:0.011

4. A detailed breakdown of the oxygenates is necessary in Figure 4 since ‘total FE of oxygenates’ is a tricky term. It is well known that CO reduction process generates a lot of acetate, which cannot be directly used as fuel additive (only ethanol and n-propanol can be directly used). More importantly, the mechanism of acetate formation is likely to be significantly different from the formation of ethanol/n-propanol and simply adding them together is not acceptable (*JACS* 141, 4191-4193). The comparison with literature shall also be based on the generation of alcohols instead of oxygenates. The ratio between oxygenates/ethylene shall also be changed to alcohols/ethylene, as shown in the authors’ previous publication, *Nat. Catal.* 1, 421-428.

We are now clear that we focus on alcohol, including only ethanol and 1-propanol, throughout the revised files and updated **Fig. 4** shown below:

Fig. 4. COR performance of Pd-doped Cu electrocatalysts. **a-d**, COR product selectivities (FE, %) and activities (partial current densities, mA cm⁻²) towards alcohols (**a, b**), ethylene (**c**) and hydrogen (**d**) productions on Cu, CuPd_{0.005}, CuPd_{0.010} and CuPd_{0.015} catalysts at various applied potentials (vs. RHE) in 1 M KOH. **e**, The peak alcohol selectivities and relevant ethylene selectivities on different catalysts at -0.66 V (Cu), -0.63 V (CuPd_{0.005}), -0.62 V (CuPd_{0.010}) and -0.53 V (CuPd_{0.015}) vs. RHE. Numbers in the red circles show the corresponding Faradaic efficiency ratio of alcohol to ethylene. Error bars are means ± SD (n = 3 replicates).

We now include detailed product distributions, by catalyst, in **Supplementary Table 7**.

Supplementary Table 7. A summary of Faradaic efficiencies for all products at different Cu catalysts.

Catalyst	Potential (V vs. RHE)	Faradaic Efficiency (%)					
		Hydrogen	Ethylene	Acetate	Ethanol	1-Propanol	Total
Cu	-0.49	17.1±2	49.3±2	5.9±0.5	11.6±2	13.8±0.5	97.7
	-0.52	13.8±1.1	50.5±1.4	5.6±0.3	13.7±1.5	10.7±1	94.3
	-0.55	10.4±0.5	51.5±0.7	6.1±0.2	14.3±0.5	10.1±0.2	92.4
	-0.58	8.5±0.3	52.8±1	8.7±1.2	15.8±1.2	9±1.2	94.8
	-0.60	8±0.2	51.4±2.5	8.1±2	17.4±1	8.6±2	93.5
	-0.62	7.8±0.5	52.9±3	9.2±1.5	17.8±1.2	8.4±0.5	96.1
	-0.64	7.3±0.7	53.4±2	9.8±1	18.6±1.2	7.8±0.3	96.9
	-0.66	6.9±0.3	54.8±1.4	9±1.3	20.1±0.8	6.9±0.6	97.7
	-0.69	6.5±0.4	55.5±2.2	11.2±2.1	19.8±2.5	5.4±0.5	98.4
-0.71	7.5±0.8	52.3±1.8	14.4±1.8	17.7±2	3.1±0.3	95	
CuPd _{0.005}	-0.47	16.3±2	33.2±2.3	8.1±0.5	17.8±1.2	13.6±1	89
	-0.51	12.6±0.5	37.7±1.5	9.2±1.5	21.8±0.6	9.5±0.5	90.8
	-0.54	11.1±1	39.5±0.7	11.5±2	23.5±1.1	9.7±0.7	95.3
	-0.57	10±1.2	40.8±2	11.5±1.2	25.9±2	8.8±0.2	97
	-0.59	9.3±0.5	41.8±2.5	9.1±1	26.3±1.4	8.1±0.5	94.6
	-0.61	9±0.3	43±1.3	10.6±0.7	28±1	7.2±0.3	97.8
	-0.63	8.9±0.2	43.9±0.5	11.6±1.3	28.5±1.2	6.9±1	99.8
	-0.65	9.2±0.5	44.4±3	12.7±1	27.4±0.5	5.6±0.4	99.3
	-0.68	9.9±1	43.7±3.2	11.4±1.4	26.2±1.2	4.8±0.2	96
-0.71	11.9±1.5	42.2±2.1	11.4±1.5	26.1±1	4.6±0.2	96.2	
CuPd _{0.010}	-0.49	14.6±2	33±1.2	7.7±1.2	23.3±1.5	14.3±2	92.9
	-0.52	12±1.2	36.2±0.5	11.3±2	25.9±0.7	10.3±1.5	95.7
	-0.54	10.6±0.5	36.2±1	12.9±0.5	27.1±1.3	8.7±0.3	95.5
	-0.56	9.2±0.8	36.6±0.6	13.1±1.3	28.7±1.2	7.6±0.2	95.2
	-0.58	9.2±1	36.9±0.4	13.9±1.1	30.1±2.1	7.4±0.3	97.5
	-0.60	8.8±0.4	36.7±0.2	14.5±0.4	31.5±0.8	6.6±0.5	98.1
	-0.62	8.5±2.5	37±1	14.9±1.2	33.2±1	6.3±0.2	99.9
	-0.64	9.5±2	36.8±0.8	18.6±1.3	29.2±1.2	5.3±0.7	99.4
	-0.67	11.6±3.2	34.6±1.5	19.2±0.7	26.1±2	4±0.3	95.5
-0.72	19±3.5	29.5±2	21±2.1	22.4±1.5	3.3±0.2	95.2	
CuPd _{0.015}	-0.51	12.8±0.5	35.1±2.3	5.7±0.5	21.2±1.2	8.4±1	83.2
	-0.53	11.5±1	35.2±1.2	8.2±1.2	25.8±1.5	5.1±0.5	85.8
	-0.58	11.6±0.2	36±2.5	11.7±2.3	25.1±1.5	4.2±1.3	88.6
	-0.60	12.7±1.2	34.8±3.2	14.4±1.4	23.7±1	2.7±0.4	88.3
	-0.65	21.3±2.3	28±4	19.2±1.2	21.4±0.7	1.5±0.5	91.4
	-0.72	29.6±3	20.3±2.3	19.5±2	16.3±2.3	0.6±0.2	86.3

We also updated the performance comparison to focus on the generation of alcohols, as tabulated in **Supplementary Table 8**.

Supplementary Table 8. Summary of alcohols electroproduction from CO₂/CO reduction tested at operating current densities > 100 mA cm⁻².

Catalyst	Reaction and cell type	Electrolyte	FE _{alcohols} (ethanol + propanol, %)	J _{alcohols} (ethanol + propanol, mA cm ⁻²)	Potential (V vs. RHE)	Reference
SA-CuPd	CORR in flow cell	1 M KOH	40	277	-0.62	This work
SA-CuPt	CORR in flow cell	1 M KOH	36	250	-0.63	
OD-Cu	CORR in flow cell	1 M KOH	32	83	-0.60	Nat. Catal. 1 , 748 – 755 (2018)
Cavity-Cu	CORR in flow cell	1 M KOH	28	45	-1.36	Nat. Catal. 1 , 946 – 951 (2018)
Cu:molecule	CO ₂ RR in flow cell	1 M KHCO ₃	41	124	-0.81	Nat. Catal. 3 , 75 – 82 (2020)
Ce(OH) _x /Cu/PTFE	CO ₂ RR in flow cell	1 M KOH	43	129	-0.70	Nat. Commun. 10 , 5814 (2019)
Bimetallic Ag/Cu	CO ₂ RR in flow cell	1 M KOH	41	103	-0.67	J. Am. Chem. Soc. 141 , 8584-8591 (2019)
Core-shell Cu ₂ S-Cu	CO ₂ RR in flow cell	1 M KOH	32	126	-0.92	Nat. Catal. 1 , 421 – 428 (2018)
CuAg alloy	CO ₂ RR in flow cell	1 M KOH	29	65	-0.64	J. Am. Chem. Soc. 140 , 5791–5797 (2018)

5. It is well known that nanoclusters may undergo agglomeration after long term electrolysis. The performance of the catalysts seemed to be evaluated over a time period of only 40 min. The stability of catalysts over >10 hours catalysis, as well as the change of catalysts, shall be presented. The stability is a critical parameter for catalyst and shall be discussed in detail.

We performed an extended runtime test to assess stability. We performed a stability test using a polytetrafluorethylene membrane with a pore size of 0.45 μm as a gas diffusion layer (Ref 4: *Science* **360**, 783-787 (2018)). This substrate was employed instead of the Sigracet 28BC GDL, to avoid flooding at long run times. By operating at a constant current density of 100 mA cm⁻², we achieved a 5-hour stable alcohol production using the CuPd_{0.010} catalyst. After a 5-hour operation, the alcohol selectivity declined due to the aggregation of Pd dopants and a partial loss of Pd loading at the Cu catalyst surface.

Supplementary Fig. 19. **a**, COR performance of the CuPd_{0.010} catalyst loaded on a polytetrafluorethylene membrane gas diffusion layer (pore size of 0.45 μm) over a course of 7 hours operation in 1 M KOH at a constant current density of 100 mA cm^{-2} . **b**, HAADF-STEM images of CuPd_{0.010} after 7 hours COR test. **c**, Elemental composition of Cu and Pd in CuPd_{0.010} before and after 7 hours COR test determined by inductively coupled plasma optical emission spectrometry (ICP-OES).

We discuss these results in the revised manuscript (**Page 10, Line 196**):

“We monitored the structure and stability of Pd dopants by carrying out a long-term COR test. A polytetrafluorethylene membrane with pore sizes of 0.45 μm was used as gas diffusion layer to avoid catholyte flooding⁴. We successfully demonstrated a stable $\text{FE}_{\text{alcohol}}$ of 40% over the course of five hours COR operation at a constant current density of 100 mA cm^{-2} (**Supplementary Fig. 19**). Alcohol selectivity decreased after five hours

due to the aggregation of Pd dopants and a partial decrease of Pd loading (**Supplementary Fig. 19b** and **19c**).”

6. The review of previous reports is out of date and shall be updated. For example, the authors quoted ‘even CO₂-CO-oxygenates tandem catalysts fail to exceed 40% oxygenate selectivity’. However, even the author’s own group has recently reported CuAg catalyst that showed 41% ethanol selectivity (JACS 141, 8584-8591) and many of similar reports exist.

We have removed the sentence and updated the performance metrics of alcohol electroproduction reported in literature within the text and in **Supplementary Table 8** shown above.

Reviewer #3

Sinton, Sargent and coworkers have produced a highly interesting paper on the reduction of CO to oxygenates. Guided by DFT calculations, they find that predict that isolated Cu atoms on Pd should enhance oxygenate selectivity. They perform experiments, that putatively verify this claim.

Although this paper is highly interesting to the broad readership of Nature Communications, it has some ambiguities, which, to my mind, prevent its publication in the current form. More specifically:

1) It is unclear what products are formed. The data in their present form lack transparency. To this end:

a. The authors should show the raw NMR/GC data for both their calibration and the actual reaction products, for each and every product. They should plot in their man text the breakdown for each product at every potential, rather than bunching all oxygenates together.

We thank the reviewer for these comments, and now provide these details in the revision.

NMR and GC raw data are provided below and are now included in SI (**Supplementary Fig. 25 and 26**).

Supplementary Fig. 25. Representative GC traces of COR gaseous products on $\text{CuPd}_{0.010}$ catalysts at -0.62 V vs. RHE in 1M KOH from thermal conductivity detector (a) and flame ionization detector (b) channels.

Supplementary Fig. 26. Representative ¹H-NMR spectra of COR liquid products on CuPd_{0.010} catalysts at -0.62 V vs. RHE in 1M KOH. DMSO (Dimethyl sulphoxide) was used as an internal for liquid products quantification.

We added the information in the revised manuscript for clarity (**Page 15, Line 317**):

“Gas products were analyzed using a gas chromatography (GC, PerkinElmer Clarus 680) equipped with a Molecular Sieve 5A capillary column and a packed Carboxen-1000 column. Argon (Linde, 99.999%) was used as the carrier gas. Hydrogen and ethylene products were quantified using a thermal conductivity detector and a flame ionization detector, respectively, equipped with a methanizer loaded in GC instrument (**Supplementary Fig. 25**). Liquid products dissolved in the catholyte were quantified using one-dimensional ¹H Nuclear magnetic resonance spectroscopy (¹H NMR) coupled with a Agilent DD2 500 spectrometer, in which a diluted Dimethyl sulfoxide (DMSO) in D₂O was used as an internal standard for the identification and quantification of liquid products (**Supplementary Fig. 26**).”

We have updated **Fig. 4** and added the selectivity breakdown of each product in **Supplementary Fig. 15** and **Supplementary Table 7**.

Fig. 4. COR performance of Pd-doped Cu electrocatalysts. a-d, COR product selectivities (FE, %) and activities (partial current densities, mA cm^{-2}) towards alcohols (**a, b**), ethylene (**c**) and hydrogen (**d**) productions on Cu, CuPd_{0.005}, CuPd_{0.010} and CuPd_{0.015} catalysts at various applied potentials (vs. RHE) in 1 M KOH. **e,** The peak alcohol selectivities and relevant ethylene selectivities on different catalysts at -0.66 V (Cu), -0.63 V (CuPd_{0.005}), -0.62 V (CuPd_{0.010}) and -0.53 V (CuPd_{0.015}) vs. RHE. Numbers in the red circles show the corresponding Faradaic efficiency ratio of alcohol to ethylene. Error bars are means \pm SD (n = 3 replicates).

Supplementary Fig. 15. COR product selectivities (FE, %) towards ethanol (a, b), 1-propanol (c) and acetate (d) productions on Cu, CuPd_{0.005}, CuPd_{0.010} and CuPd_{0.015} catalysts at various applied potentials (vs. RHE) in 1 M KOH.

Supplementary Table 7. A summary of Faradaic efficiencies for all products at different Cu catalysts.

Catalyst	Potential (V vs. RHE)	Faradaic Efficiency (%)					
		Hydrogen	Ethylene	Acetate	Ethanol	1-Propanol	Total
Cu	-0.49	17.1±2	49.3±2	5.9±0.5	11.6±2	13.8±0.5	97.7
	-0.52	13.8±1.1	50.5±1.4	5.6±0.3	13.7±1.5	10.7±1	94.3
	-0.55	10.4±0.5	51.5±0.7	6.1±0.2	14.3±0.5	10.1±0.2	92.4
	-0.58	8.5±0.3	52.8±1	8.7±1.2	15.8±1.2	9±1.2	94.8
	-0.60	8±0.2	51.4±2.5	8.1±2	17.4±1	8.6±2	93.5
	-0.62	7.8±0.5	52.9±3	9.2±1.5	17.8±1.2	8.4±0.5	96.1
	-0.64	7.3±0.7	53.4±2	9.8±1	18.6±1.2	7.8±0.3	96.9
	-0.66	6.9±0.3	54.8±1.4	9±1.3	20.1±0.8	6.9±0.6	97.7
	-0.69	6.5±0.4	55.5±2.2	11.2±2.1	19.8±2.5	5.4±0.5	98.4
-0.71	7.5±0.8	52.3±1.8	14.4±1.8	17.7±2	3.1±0.3	95	
CuPd _{0.005}	-0.47	16.3±2	33.2±2.3	8.1±0.5	17.8±1.2	13.6±1	89
	-0.51	12.6±0.5	37.7±1.5	9.2±1.5	21.8±0.6	9.5±0.5	90.8
	-0.54	11.1±1	39.5±0.7	11.5±2	23.5±1.1	9.7±0.7	95.3
	-0.57	10±1.2	40.8±2	11.5±1.2	25.9±2	8.8±0.2	97
	-0.59	9.3±0.5	41.8±2.5	9.1±1	26.3±1.4	8.1±0.5	94.6
	-0.61	9±0.3	43±1.3	10.6±0.7	28±1	7.2±0.3	97.8
	-0.63	8.9±0.2	43.9±0.5	11.6±1.3	28.5±1.2	6.9±1	99.8
	-0.65	9.2±0.5	44.4±3	12.7±1	27.4±0.5	5.6±0.4	99.3
	-0.68	9.9±1	43.7±3.2	11.4±1.4	26.2±1.2	4.8±0.2	96
-0.71	11.9±1.5	42.2±2.1	11.4±1.5	26.1±1	4.6±0.2	96.2	
CuPd _{0.010}	-0.49	14.6±2	33±1.2	7.7±1.2	23.3±1.5	14.3±2	92.9
	-0.52	12±1.2	36.2±0.5	11.3±2	25.9±0.7	10.3±1.5	95.7
	-0.54	10.6±0.5	36.2±1	12.9±0.5	27.1±1.3	8.7±0.3	95.5
	-0.56	9.2±0.8	36.6±0.6	13.1±1.3	28.7±1.2	7.6±0.2	95.2
	-0.58	9.2±1	36.9±0.4	13.9±1.1	30.1±2.1	7.4±0.3	97.5
	-0.60	8.8±0.4	36.7±0.2	14.5±0.4	31.5±0.8	6.6±0.5	98.1
	-0.62	8.5±2.5	37±1	14.9±1.2	33.2±1	6.3±0.2	99.9
-0.64	9.5±2	36.8±0.8	18.6±1.3	29.2±1.2	5.3±0.7	99.4	

	-0.67	11.6±3.2	34.6±1.5	19.2±0.7	26.1±2	4±0.3	95.5
	-0.72	19±3.5	29.5±2	21±2.1	22.4±1.5	3.3±0.2	95.2
CuPd _{0.015}	-0.51	12.8±0.5	35.1±2.3	5.7±0.5	21.2±1.2	8.4±1	83.2
	-0.53	11.5±1	35.2±1.2	8.2±1.2	25.8±1.5	5.1±0.5	85.8
	-0.58	11.6±0.2	36±2.5	11.7±2.3	25.1±1.5	4.2±1.3	88.6
	-0.60	12.7±1.2	34.8±3.2	14.4±1.4	23.7±1	2.7±0.4	88.3
	-0.65	21.3±2.3	28±4	19.2±1.2	21.4±0.7	1.5±0.5	91.4
	-0.72	29.6±3	20.3±2.3	19.5±2	16.3±2.3	0.6±0.2	86.3

We included the information in the revised manuscript (**Page 8, Line 167**):

“Selectivities (FE, %) for individual products from COR using different Cu catalysts are provided in **Supplementary Fig. 15** and **Supplementary Table 7**, and the corresponding product activities (current density, mA cm⁻²) are shown in **Fig. 4** and **Supplementary Fig. 16**.”

b. They should plot the partial current densities for each and every product at each potential.

We have now plotted the partial current densities for each and every product at each potential in **Supplementary Fig. 16**.

Supplementary Fig. 16. COR product activities (partial current densities, mA cm⁻²) towards ethanol (a), 1-propanol (b), ethylene (c), acetate (d) and hydrogen (e) productions on Cu, CuPd_{0.005}, CuPd_{0.010} and CuPd_{0.015} catalysts at various applied potentials (vs. RHE) in 1 M KOH.

c. They should also plot the unbalanced charge for at each potential

We further quantify the missing current as below:

Figure R2. Missing current densities on Cu, CuPd_{0.005}, CuPd_{0.010} and CuPd_{0.015} catalysts at various applied potentials (vs. RHE) in 1 M KOH.

2) *The authors should calibrate the reference electrode experimentally using a real RHE.*

We have calibrated our Ag/AgCl/1 mg/kg KCl reference electrode as detailed below:

Figure R3. Calibration of Ag/AgCl/1 mg/kg KCl reference electrode ($E^0 = 0.235$ V at 25 °C). Test was performed in ultrapure H₂-saturated 1 M HClO₄ solution (pH = 0), the onset potential of hydrogen evolution reaction on polished Pt foil was determined to be 0.240 – 0.235 = 0.005 V \approx 0 V, indicating the accuracy of Ag/AgCl/1 mg/kg KCl reference electrode used in this work.

3) They should also correct for Ohmic losses and state exactly how they measured Ohmic losses.

We added the ohmic loss correction information into the revised manuscript (**Page 14, Line 308**):

“Potentials reported in this work were calculated to the reversible hydrogen electrode (RHE) reference scale using $E_{\text{RHE}} = (E_{\text{Ag/AgCl}})_{iR} + 0.235 \text{ V} + 0.059 \times \text{pH}$. The ohmic drop correction (i.e. iR compensation: i is the applied current and R is the cell resistance) was conducted using $(E_{\text{Ag/AgCl}})_{iR} = E_{\text{Ag/AgCl}} - 0.85 \times i \times R$, in which $E_{\text{Ag/AgCl}}$ was the applied potential before iR compensation and a R value of 3.4Ω was determined by performing an electrochemical impedance spectroscopy measurement using an Autolab PGSTAT302N electrochemical workstation coupled with a FRA32M module. A factor of 0.85 is applied in iR compensation during flow cell operation due to a low resistivity of 1 M KOH electrolyte which holds a relatively low voltage drop over the electrolyte.¹⁸”

4) The authors should provide more convincing evidence to explain why their synthesis method should favour single atom catalysts.

Cu has a lower reduction potential (Cu^{2+}/Cu , $E^0 = +0.340 \text{ V}$ vs. NHE) than Pd (Pd^{2+}/Pd , $E^0 = +0.915 \text{ V}$ vs. NHE) and Pt ($[\text{PtCl}_6]^{2-}/\text{Pt}$, $E^0 = +1.435 \text{ V}$ vs. NHE). The energetically favourable galvanic replacement of Cu with Pd/Pt proceeds by:

The reduction of $\text{Pd}^{2+}/[\text{PtCl}_6]^{2-}$ by Cu can be facilitated by ultrasound as sonochemical effect enhances reaction rates at ambient conditions (Ref 27: *Phys. Chem. Chem. Phys.* **15**, 12187-12196 (2013)). In the case of adding a small amount of Pd/Pt precursors in the bulk Cu solutions, ultrasound waves consisting of compression and rarefaction cycles not only improve the reaction rate of galvanic replacement but also inhibit the aggregation of Pd/Pt atoms, in agreement of prior works synthesizing Pd/Pt doped Cu catalysts via ultrasound-assisted galvanic replacement (Ref 27: *Phys. Chem. Chem. Phys.* **15**, 12187-12196 (2013); Ref 42: *Nat. Commun.* **10**, 5812 (2019)).

We clarified these aspects in the revised *Methods* section (**Page 12, Line 251**):

“The energetically favourable galvanic replacement of Cu with Pd/Pt proceeds by $\text{Cu} + \text{Pd}^{2+} \rightarrow \text{Cu}^{2+} + \text{Pd}$ or $2\text{Cu} + [\text{PtCl}_6]^{2-} \rightarrow 2\text{Cu}^{2+} + \text{Pt} + 6\text{Cl}^-$. In the case of adding a small amount of Pd/Pt precursor in the bulk Cu solutions, ultrasound waves consisting of compression and rarefaction cycles not only improve the reaction rate of galvanic replacement but also inhibit the aggregation of Pd/Pt atoms, in agreement with prior works^{27, 42}.”

5) All data are performed *ex-situ*, meaning that the surface atoms will inevitably be oxidised by air transfer and hence unrepresentative of the catalytically active state. Soriaga and coworkers have demonstrated how mobile Cu atoms are under CO₂ reduction conditions. For instance, reference 9 shows that a catalyst that forms single atom Cu reversibly aggregates under reaction conditions and then reverts to the single atom form *ex-situ*. To this end, it would be far more convincing if the authors would provide *in-situ* spectroscopic data to demonstrate that they have single atom catalysts, either based on FTIR (see works by Peter Stair or Maria Stephanopoulos for gas phase reactions) or XAS (reference 9 of current text).

All Cu K-edge and Pt L₃-edge XAS spectra in this work were indeed performed *in operando*, at the 9BM beamline of the Advanced Photon Source (9BM@APS, <https://www.aps.anl.gov/Spectroscopy/Beamlines/9-BM>). We clarify this aspect in the manuscript and reference details of the operando XAS cell design and testing in our previous work (Ref 16: *Nat. Commun.* **9**, 4614 (2018)).

Since the absorption energy of Pd K-edge (~24350 eV) exceeds the detection energy range of 9BM beamline (2.1 – 24 keV), we conducted the Pd K-edge measurements at 20BM beamline of APS with a detection energy range of 2.7 – 30 keV (<https://www.aps.anl.gov/Spectroscopy/Beamlines/20-BM>). However, the sample testing stage at 20BM was in an open system and could not accommodate CO gas for operando COR testing. In this case we performed instead *ex-situ* Pd K-edge testing. Samples after COR were washed with DI water, dried using N₂ gas and quickly sealed with Kapton tape, no oxidation of Pd was observed in our XAS spectra, suggesting the presence of metallic active Pd sites during COR.

We further assessed the Pd structure by carrying out a long-term COR test using the Pd-doped Cu catalysts (**Supplementary Fig. 19**). After a 7-hour COR operation, some Pd atoms had aggregated, reducing alcohol selectivity and increasing H₂ production. These results indicate importance of atomic-level Pd dopants in assisting alcohol production from COR, consistent with our DFT calculations.

Supplementary Fig. 19. **a**, COR performance of the CuPd_{0.010} catalyst loaded on a polytetrafluorethylene membrane gas diffusion layer (pore size of 0.45 μm) over a course of 7 hours operation in 1 M KOH at a constant current density of 100 mA cm^{-2} . **b**, HAADF-STEM images of CuPd_{0.010} after 7 hours COR test. **c**, Elemental composition of Cu and Pd in CuPd_{0.010} before and after 7 hours COR test determined by inductively coupled plasma optical emission spectrometry (ICP-OES).

We corrected the following information in revised manuscript for clarity:

Page 10, Line 196: “We monitored the structure and stability of Pd dopants by carrying out a long-term COR test. A polytetrafluorethylene membrane with pore sizes of 0.45 μm was used as gas diffusion layer to avoid catholyte flooding⁴. We successfully demonstrated a stable $\text{FE}_{\text{alcohol}}$ of 40% over a course of five hours COR operation at a constant current density of 100 mA cm^{-2} (**Supplementary Fig. 19**). Alcohol selectivity decreased after five hours due to the

aggregation of Pd dopants and a partial decrease of Pd loading (**Supplementary Fig. 19b and 19c**).”

Page 13, Line 280: “Operando XAS measurements at the Cu K-edge and Pt L₃-edge were carried out at the 9BM beamline of Advanced Photon Source (APS, Argonne National Laboratory, Lemont, Illinois) in a flow cell¹⁶. Ex-situ Pd K-edge experiments were conducted at the 20BM beamline of APS by sealing samples in Kapton tape right after the COR testing. Fluorescence yield was recorded using solid state detectors at both the 20BM and 9BM beamlines.”

Page 23, Line 481: “Cu K-edge XAS was performed in operando mode during COR at -0.62V vs. RHE in 1 M KOH, whereas ex-situ Pd K-edge was conducted after COR at -0.62V vs. RHE in 1 M KOH.”

If the authors are able to convincingly address my concerns above, then their work might be suitable for publication.

The revised manuscript is significantly improved thanks to the insightful comments of Reviewer #3.

Reviewers' Comments:

Reviewer #1:

Remarks to the Author:

The authors have more than adequately addressed each comment and the efforts they have made are commendable. I recommend the manuscript is published, however, now that they have fully established the Pd content in their samples through ICP, this should be included throughout the manuscript instead of the SampleID. This will more accurately represent the composition of each sample for their discussion.

Reviewer #2:

Remarks to the Author:

The authors have addressed all of my concerns very well and I would be happy to see this manuscript to be published in the Nature Communications.

I have one minor comment on the answer to my question No. 1, which does not require additional revision but would like to ask the authors to consider. The authors mentioned that the stoichiometry of hydrogen for ethylene and ethanol production is the same. But please note that this is only equation! In real case, once we consider the multiple steps of the reaction, the stoichiometry could be different. Additionally, I think that 'hydrogen' is from water, instead of proton directly. H₂O concentration is 56 mol/L while the proton concentration is only 10⁻¹⁴ mol/L in 1 M KOH, which is nearly 10¹⁶ lower. The preferred equations, in my opinion, shall be:

Reviewer #3:

Remarks to the Author:

The authors have adequately addressed my concerns and those of the other reviewers. This manuscript should now be published.

Manuscript ID: NCOMMS-19-2396360A

“Enhanced multi-carbon alcohol electroproduction from CO via modulated hydrogen adsorption”

Reviewer 1

The authors have more than adequately addressed each comment and the efforts they have made are commendable. I recommend the manuscript is published, however, now that they have fully established the Pd content in their samples through ICP, this should be included throughout the manuscript instead of the SampleID. This will more accurately represent the composition of each sample for their discussion.

We thank Review #1 for this feedback. We have renamed samples with their corresponding Pd:Cu ratios as determined by ICP-OES, and applied this convention throughout the revised manuscript.

Reviewer #2

*The authors have addressed all of my concerns very well and I would be happy to see this manuscript to be published in the Nature Communications. I have one minor comment on the answer to my question No. 1, which does not require additional revision but would like to ask the authors to consider. The authors mentioned that the stoichiometry of hydrogen for ethylene and ethanol production is the same. But please note that this is only equation! In real case, once we consider the multiple steps of the reaction, the stoichiometry could be different. Additionally, I think that 'hydrogen' is from water, instead of proton directly. H₂O concentration is 56 mol/L while the proton concentration is only 10⁻¹⁴ mol/L in 1 M KOH, which is nearly 10¹⁶ lower. The preferred equations, in my opinion, shall be:
2CO + 7H₂O + 8e⁻ = C₂H₅OH + 8OH⁻
2CO + 6H₂O + 8e⁻ = C₂H₄ + 8OH⁻*

We have changed the two equations as required:

We also have replaced H⁺ with H₂O as proton source in revised **Supplementary Equations 1-4 (Supplementary Note 1)**:

“The hydrogenation of HOCCH at the Cu surface using either a pre-adsorbed H or H from H₂O was simulated according to **Supplementary Equation 1** or **2**:

where the * represent the adsorption site. Similarly, the dehydroxylation process was simulated by **Supplementary Equation 3**:

the adsorption of H on different surfaces was calculated based on **Supplementary Equation 4**:

»

Reviewer #3

*The authors have adequately addressed my concerns and those of the other reviewers.
This*

We thank the reviewer.